# Provenance-Enabled Multi-View Diabetic Retinopathy Diagnosis Through Interpretable Process Mining

## Abstract

Diabetic retinopathy (DR) is a leading cause of blindness among individuals with diabetes. Although the existing deep learning models have demonstrated potential in DR diagnosis, they still lack full-process interpretability. Specifically, these models suffer from three key challenges: reliance on single-source inputs, opaque and untraceable reasoning processes, and the absence of a mechanism for result verification. To meet the requirements of the medical scenario for a trustworthy diagnostic model, we propose a provenance-enabled concept-based framework for multi-view DR diagnostic (ProConMV). This work integrates DR lesion masks, clinical text and multi-view data, utilizing multimodal prompt analysis and visual-text concept interaction to learn the interpretable multi-source input. During the reasoning stage, the proposed framework introduces lesion concepts for causal reasoning chains combining clinical guidelines, and adds doctor intervention for human-machine collaboration. For dynamic fusion decision and verification in multi-view DR diagnosis, we derive via generalization theory that incorporating each view's lesion concept uncertainty and grading uncertainty reduces the generalization error upper bound. Accordingly, we design a dual uncertainty-aware module to enable provenance-based verification, ultimately enabling verifiable analysis of DR diagnostic results. Extensive experiments conducted on two public multi-view DR datasets demonstrate the effectiveness of our method.

## 1 Introduction

Diabetic retinopathy (DR) is a major cause of blindness among diabetic patients (Federation, 2021), posing a visual health error to the global working-age population. International DR severity is diagnosed by lesions like microaneurysms (MA), hemorrhage (HE), and exudation (EX), and classified into five grades (Grade 0-4): normal, mild, moderate, severe, and Proliferative Diabetic Retinopathy (PDR) (Wilkinson et al., 2003). With the development of artificial intelligence technology, traditional deep learning models (Liu et al., 2022a; 2024a) have demonstrated excellent performance in DR grading tasks, capable of quickly processing large amounts of images and providing grading results. However, their inherent limitations in practical application have gradually become bottlenecks in bridging the gap between AI technology and real-world medical needs.

A critical examination of existing DR diagnostic models (as in Section A.1 of appendix) reveals three core challenges that undermine their credibility and usability in clinical settings (Lin et al., 2025), as illustrated in Fig.1. First, single-source input limitations persist: most models rely solely on monomodal data and fail to integrate complementary information from lesion morphology and clinical text. Moreover, training on single-view databases (Decenciere et al., 2014; EyePACS, 2015) means the field of view (FOV) of input images covers only 20% of the observable fundus, increasing the error of missing critical pathological features. Second, "black-box" reasoning processes lack medical interpretability (Huang et al., 2024): the internal calculations of traditional models are opaque, and they cannot map image features to diagnostic results via clinically understandable logic. Third, insufficient result verification mechanisms (Luo et al., 2025): existing methods generally lack uncertainty quantification and traceable validation, making it impossible to assess the reliability of diagnostic outputs. This deficiency is particularly problematic in medical scenarios, where unreliable results may lead to misdiagnosis, missed diagnosis, or inappropriate clinical interventions.

Figure 1: Our model with an interpretable process compared with traditional DR diagnosis models.

To address these critical issues and alleviate the credibility dilemma of DR diagnosis models in clinical practice, this study proposes a full-process interpretable framework for DR diagnosis, encompassing multi-source input fusion, interpretable causal reasoning, and verifiable result evaluation. Specifically, we integrate fundus image lesion masks, structured clinical texts, and multi-view fundus data to construct a rich input space. The proposed Hilbert RWKV encodes spatial features of images for precise lesion localization, while a large language model (LLM)-based text encoder (Achiam et al., 2023) extracts lesion-related semantic information from clinical texts, with cross-modal interaction enabled by a Visual-Text RWKV (VT-RWKV) module. For reasoning, we introduce lesion concepts (Wen et al., 2024) as intermediate units aligned with clinical guidelines. And incorporate real-time doctor intervention to build a human-machine collaborative causal reasoning chain, transforming "input-output" mapping into physician-understandable pathological logic.

Furthermore, multi-view fusion decision-making is crucial for the comprehensive DR diagnosis. However, due to varying cooperation among different patients during fundus examinations, the captured multi-view fundus images exhibit various variations. Most existing multi-view fusion methods (Hu et al., 2025) lack theoretical guarantees, which can lead to one-sided and inaccurate diagnostic results. To achieve reliable dynamic fusion, we demonstrate for the first time in a multi-view concept-based model that, from the perspective of generalization theory, when fusion weights are negatively correlated with both concept loss and grading loss, the upper bound of the generalization error for decision fusion will be reduced and outperforms that of static fusion methods. Meanwhile, the concept uncertainty and grading uncertainty of each view related to the decision are traceable, enabling verifiable analysis of DR diagnostic results. The main contributions of this full-process interpretable DR diagnosis framework are summarized as follows:

- The multimodal input mechanism is proposed to integrate DR lesion masks, clinical text, and multi-view data. Leveraging Hilbert RWKV encoding of image features and textual concept encoder extraction of text features to achieve cross-modal interaction, a semantically rich interpretable input foundation is provided for reasoning.

- A causal reasoning chain combining lesion concepts and clinical guidelines is constructed, with the simultaneous introduction of a doctor intervention link to form a human-machine collaborative reasoning mode, effectively solving the problem of opaque and untraceable reasoning processes in traditional models.

- In the dynamic fusion decision, we derive for the first time from the perspective of generalization that incorporating the lesion concept uncertainty and the grading uncertainty of each view can reduce the generalization error upper bound. Then, we design a dual uncertainty-aware module to realize provenance-enabled verification of diagnostic results.

## 2 METHOD

This framework takes the fusion of multi-source clinical data as its input foundation, uses medically logical causal reasoning as its core link, and employs a dual uncertainty-aware mechanism as its result guarantee.

Figure 2: The framework of our proposed ProConMV model has three parts: multi-source input for the enhancement of interpretable features, visual-text concepts integration for causal reasoning, and provenance-enabled diagnosis using the dual uncertainty-aware module.

## 2.1 MULTI-VIEW CONCEPT REPRESENTATION LEARNING

Some studies (Xu et al., 2021; Shamshad et al., 2023) have demonstrated that existing Transformer-based multi-view methods (Xu et al., 2024; Gu et al., 2024) are less effective at fine-grained local concept perception, while incurring large parameter overhead and prolonged inference times. To capture multi-view fine-grained lesion concept features, we propose an RWKV-based backbone equipped with multi-directional Hilbert attention mechanism, which preserves linear complexity while ensuring continuity in fundus local representation learning. Specifically, the backbone first utilizes a stem (comprising two convolutional layers and downsampling) to extract shallow features for each view. Then, it optimizes the deep features using two Hilbert RWKV Blocks.

### 2.1.1 HILBERT RWKV BLOCK

This block mainly consists of two components: Hilbert spatial-mix and channel-mix. The spatial mixing is the core, while the channel mixing serves as a feed-forward network (FFN) to enhance channel features. Given the fundus representation of the $v$-th view $\mathbf{x}^{(v)} \in \mathbb{R}^{h \times w \times d}$, the block first transforms it into $p \times p$ patches, which are then projected into visual tokens of shape $\frac{hw}{p^2} \times d$. These tokens $\overline{\mathbf{x}}^{(v)}$ are fed into the Hilbert spatial-mix module. Similar to Vision-RWKV (Duan et al., 2025), we adopt the quad-directional token shift (Q-Shift) operation along with three parallel linear layers to obtain the matrices $\mathbf{R}_s, \mathbf{K}_s, \mathbf{V}_s \in \mathbb{R}^{\frac{hw}{p^2} \times d}$:

$$\mathbf{R}_s = \text{Q-Shift}_R(\overline{\mathbf{x}}^{(v)})W_R, \quad \mathbf{K}_s = \text{Q-Shift}_K(\overline{\mathbf{x}}^{(v)})W_K, \quad \mathbf{V}_s = \text{Q-Shift}_V(\overline{\mathbf{x}}^{(v)})W_V. \quad (1)$$

This Q-Shift operation enhances the attention mechanism by allowing tokens to shift and perform linear interpolation with neighboring tokens, thereby improving the receptive field of each token without increasing computational complexity. The following formula holds:

$$\text{Q-Shift}_{(*)}(\overline{\mathbf{x}}^{(v)}) = \overline{\mathbf{x}}^{(v)} + (1 - \mu(*))\overline{\mathbf{x}}'^{(v)}, \quad \overline{\mathbf{x}}'^{(v)}[a, b] =$$
$$\text{Concat}(\overline{\mathbf{x}}^{(v)}[a-1, b, 0{:}\tfrac{d}{4}], \ \overline{\mathbf{x}}^{(v)}[a+1, b, \tfrac{d}{4}{:}\tfrac{d}{2}], \ \overline{\mathbf{x}}^{(v)}[a, b-1, \tfrac{d}{2}{:}\tfrac{3d}{4}], \ \overline{\mathbf{x}}^{(v)}[a, b+1, \tfrac{3d}{4}{:}d]), \quad (2)$$

where the subscript $(*) \in \{R, K, V\}$ represents the interpolation of $\overline{\mathbf{x}}^{(v)}$ and $\overline{\mathbf{x}}'^{(v)}$, controlled by the learnable vector $\mu(*)$. Subsequently, we design a novel linear attention mechanism with local continuity perception, Hilbert-WKV$(\mathbf{K}_s, \mathbf{V}_s)$, and a gating function $\sigma(\mathbf{R}_s)$ to obtain the output of the Hilbert spatial mixing module $\mathbf{O}_s$, as shown in the figure:

$$\mathbf{O}_s = \text{LN}\big(\sigma(\mathbf{R}_s) \odot \text{Hilbert-WKV}(\mathbf{K}_s, \mathbf{V}_s)W_{O_s}\big). \quad (3)$$

Here, $\sigma$ represents the sigmoid function, $\odot$ denotes element-wise multiplication, and LN refers to layer normalization. To achieve channel feature fusion, $\mathbf{O}_s$ is passed into the channel-mix module.

$\mathbf{R}_c, \mathbf{K}_c, \mathbf{V}_c \in \mathbb{R}^{\frac{hw}{p^2} \times d}$ are obtained similarly to spatial-mix by $\mathbf{O}_s$. In the channel-mix module, $\mathbf{V}_c$ is the linear projection of $\mathbf{K}_c$ after applying the activation function SquaredReLU, controlled by a gating mechanism $\sigma(\mathbf{R}_c)$. The output $\mathbf{O}_c$ is the linear projection of the resulting value:

$$\mathbf{O}_c = \sigma(\mathbf{R}_c) \odot (\text{SquaredReLU}(\mathbf{K}_c)W_V) W_{O_c}. \tag{4}$$

### 2.1.2 HILBERT-WKV ATTENTION MECHANISM

Inspired by the filling curve (Chen et al., 2023) and the bidirectional attention mechanism Bi-WKV (Duan et al., 2025), we design the Hilbert-WKV, a multi-directional attention mechanism grounded in the Hilbert curve. Our proposed Hilbert-WKV has two advantages in multi-view fundus representation learning, as shown in Fig. 2, it preserves the continuity of token arrangement, and the local scanning characteristic of the Hilbert curve window outperforms the default strip scanning.

Specifically, after dividing into $\frac{hw}{p^2}$ tokens of size $p \times p$, the arrangement order of the tokens is determined based on the 2D Hilbert curve:

$$H_n(a,b) = \begin{cases} 4 \cdot H_{n-1}(b,a) & (a,b) \in Q_0, \\ 4 \cdot H_{n-1}(a,b) + 4^{n-1} & (a,b) \in Q_1, \\ 4 \cdot H_{n-1}(a,b) + 2 \cdot 4^n & (a,b) \in Q_2, \\ 4 \cdot H_{n-1}(N-1-b, N-1-a) + 3 \cdot 4^{n-1} & (a,b) \in Q_3. \end{cases} \tag{5}$$

Here, $H_n(a,b)$ represents the Hilbert sequence position of the token located at $(a,b)$, with $N = 2^n = \frac{1}{p}\sqrt{hw}$ and $H_n(0,0) = 0$. $Q_0$ to $Q_3$ represent the four quadrants formed by dividing the area of $N/2$ into four sections: $Q_0$ (lower-left), $Q_1$ (upper-left), $Q_2$ (upper-right), and $Q_3$ (lower-right). We denote the Hilbert Transform as $\eta$ and its inverse as $\eta^{-1}$. The proposed Hilbert-WKV attention mechanism constructs attention mechanisms with vertical and horizontal direction priorities:

$$\text{Hilbert-WKV}(\mathbf{K}_s, \mathbf{V}_s) = \underbrace{\eta^{-1}(\text{Bi-WKV}(\bar{\mathbf{K}}_s, \bar{\mathbf{V}}_s))}_{Vertical\ Attention} + \underbrace{\eta^{-1}(\text{Bi-WKV}(\bar{\mathbf{K}}_s^{\mathsf{T}}, \bar{\mathbf{V}}_s^{\mathsf{T}}))^{\mathsf{T}}}_{Horizontal\ Attention},$$
$$\text{where} \quad \bar{\mathbf{K}}_s = \eta(\mathbf{K}_s), \ \bar{\mathbf{V}}_s = \eta(\mathbf{V}_s), \tag{6}$$

where $\mathsf{T}$ is the transpose. The Bi-WKV attention calculation for the $t$-th token is formulated as follows:

$$wkv_t = \text{Bi-WKV}(\mathbf{K}_s, \mathbf{V}_s)_t = \frac{\sum_{i=0,i\neq t}^{T-1} e^{-(|t-i|-1)/T \cdot w + k_i} v + e^{u+k_t} v_t}{\sum_{i=0,i\neq t}^{T-1} e^{-(|t-i|-1)/T \cdot w + k_i} + e^{u+k_t}}, \tag{7}$$

where, $T = \frac{hw}{p^2}$ represents the total number of tokens. $w$ and $u$ are two $D$-dimensional learnable vectors representing channel-wise spatial decay and the current token, respectively. $k_t$ and $v_t$ denote the $t$-th feature of $\bar{\mathbf{K}}_s$ and $\mathbf{V}_s$. Compared to the self-attention, the Hilbert-WKV attention achieves linear computational complexity $O(n \times T \times D)$, where $n$ is a constant.

### 2.1.3 VISUAL CONCEPT ENCODER

Following the shared backbone processing, each view obtains its latent representation $\overline{\mathbf{h}}^{(v)} \in \mathbb{R}^{n_h}$. Our model then feeds $\overline{\mathbf{h}}^{(v)}$ into a concept-specific fully connected layer, which learn the lesion concept embedding in $\mathbb{R}^{n_z}$, namely $\mathbf{z}_j^{(v)} = \sigma(W_j\overline{\mathbf{h}}^{(v)} + b_j)$. Here, $\mathbf{z}_v^j$ denotes the $j$-th concept embedding in the $i$-th view, while $\sigma$, $W_j$, and $b_j$ correspond to the LeakyReLU activation function, weight paremeters, and bias term of the $j$-th concept layer, which are shared across all views. In this way, the fundus visual feature is mapped into lesion concept representations for each view.

### 2.1.4 TEXTUAL CONCEPT ENCODER

We use GPT-4 (Achiam et al., 2023) to obtain medical knowledge descriptions for each DR lesion concept, focusing on their characteristics and occurrence stages. This description text is a curated knowledge base for retinal diagnosis, which provides a unified textual description for all samples as a shared semantic anchor point. The text is fed into a frozen text encoder text-embedding-3-large (TE3) to generate the textual concept embedding $\mathbf{t}_j \in \mathbb{R}^{n_t}$, where $j$ denotes the $j$-th concept.

## 2.2 MULTI-VIEW VISUAL-TEXT CONCEPT INTEGRATION

To efficiently align multi-view lesion concept representations with human clinical knowledge and improve reasoning interpretability, we propose a visual-text RWKV (VT-RWKV) block, a multi-model driven concept enhancement method based on RWKV.

Specially, for view $v$, our model considers the concatenated representation of the concept visual embedding $\mathbf{z}^{(v)} = [\mathbf{z}_1^{(v)}, \mathbf{z}_1^{(v)}, \ldots, \mathbf{z}_M^{(v)}]$ and its corresponding textual embedding $\mathbf{t} = [\mathbf{t}_1, \mathbf{t}_2, \ldots, \mathbf{t}_M]$ as input, where $M$ denotes the number of lesion concepts. The multi-modal embeddings are then projected through three parallel linear layers to obtain the matrices $\mathbf{R}_{con}, \mathbf{W}_{con}, \mathbf{K}_{con} \in \mathbb{R}^{m \times n_z}$:

$$\mathbf{R}_{con} = W_r \mathbf{z}^{(v)}, \quad \mathbf{K}_{con} = W_k \mathbf{t}, \quad \mathbf{V}_{con} = W_v \mathbf{t}, \tag{8}$$

where $W_r$, $W_k$, and $W_v$ are learnable parameters. Here, the VT-RWKV operator improves concept visual representations by fusing them with aligned textual features. The key and value matrices $\mathbf{K}_{con}$ and $\mathbf{V}_{con}$, computed from $\mathbf{t}$, are fed into a linear complexity bidirectional attention module, Bi-WKV, to obtain the attention output $wkv \in \mathbb{R}^{M \times n_z}$. Meanwhile, the visual embedding $\mathbf{z}^{(v)}$ generates a gating matrix $\sigma(\mathbf{R}_{con})$, which modulates the attention output. The enhanced concept representation $\bar{\mathbf{z}}^{(v)}$ is computed as:

$$\bar{\mathbf{z}}^{(v)} = (\sigma(\mathbf{R}_{con}) \odot cwkv) W_z, \quad cwkv = \text{Bi-WKV}(\mathbf{K}_{con}, \mathbf{V}_{con}), \tag{9}$$

where $W_z$ is a learnable projection matrix, $\sigma$ denotes the sigmoid function, and $\odot$ represents element-wise multiplication. Through this fusion, the model obtains each view's lesion concept embeddings that are aligned with both visual information and diagnostic knowledge, thereby enhancing the interpretability and predictive accuracy of the concepts.

In reasoning, the view-shared concept decoder $C$ transforms the enhanced concept representation of each view into its corresponding concept predictions, which are then passed to the grade decoder $G$ to produce the final grading result for that view. For view $v$, the procedure can be derived as:

$$\hat{\mathbf{c}}^{(v)} = C(\bar{\mathbf{z}}^{(v)}) \in \mathbb{R}^M, \quad \hat{\mathbf{y}}^{(v)} = G(\hat{\mathbf{c}}^{(v)}) \in \mathbb{R}^K. \tag{10}$$

Here, $\hat{\mathbf{c}}^{(v)}$ denotes the concept prediction vector of view $v$ with $M$ concepts, and $\hat{\mathbf{y}}^{(v)}$ represents the corresponding grading vector with $K$ DR grades. In this way, the model completes the entire process from input to concept analysis and finally to grading output for each view, i.e., $\mathbf{x}^{(v)} \rightarrow \hat{\mathbf{c}}^{(v)} \rightarrow \hat{\mathbf{y}}^{(v)}$.

## 2.3 DUAL UNCERTAINTY-AWARE INTERPRETABLE MULTI-VIEW DR DIAGNOSIS

In the traditional interpretable concept reasoning pipeline, we find that the final grading prediction cannot fully capture the reliability of each view. This limitation arises because the lesion concepts, which serve as the input to the $\mathbf{c} \rightarrow \mathbf{g}$ stage, play a critical role in determining both the interpretability and the accuracy of the reasoning process for each view. Thus, we propose a dynamic fusion method with dual uncertainty awareness in concept prediction and grading.

### 2.3.1 GENERALIZATION THEORY IN MULTI-VIEW CONCEPT-BASED MODELS

We integrate the generalization theory into the multi-view concept-based framework. This enables us to formalize the reasoning process, analyze grading prediction loss, and validate dynamic weight design, as detailed in the following setting and derivation.

**Setting.** In conjunction with Equation (10), we define $\mathbf{c}^{(v)}$ and $\mathbf{y}^{(v)}$ as the concept label and the grading (class prediction) label of view $v$, respectively. According to the reasoning pipeline $\mathbf{x}^{(v)} \rightarrow \hat{\mathbf{c}}^{(v)} \rightarrow \hat{\mathbf{y}}^{(v)}$, The view-shared concept predictor $C$ and grading predictor $G$ are specified in the hypothesis spaces $\mathcal{C}$ and $\mathcal{G}$. The final prediction of the late-fusion multi-view method is formulated as $\hat{\mathbf{y}} = \sum_{v=1}^{V} w_v \hat{\mathbf{y}}^{(v)}$, where $w_v \in (0, 1)$ denotes the fusion weight of view $v$, satisfying $\sum_{v=1}^{V} w_v = 1$. Unlike the static fusion weight $w_v^s$, dynamic weight $w_v^d$ is dependent on the input. To provide a provable dynamic weight design for multi-view concept-based models, we introduce generalization theory. The generalization error of grading (classification) prediction in multi-view concept-based models $L_y$ can be expressed as:

$$L_y := \mathbb{E}_{(\mathbf{x}^{(1:V)}, \mathbf{c}^{(1:V)}, \mathbf{y}) \sim \mathcal{D}} \left[ \ell_y \left( \sum_{v=1}^{V} w_v \, G(\hat{\mathbf{c}}^{(v)}), \mathbf{y} \right) \right], \tag{11}$$

where $\mathbb{E}$ is the expectation, $\mathcal{D}$ is the unknown joint distribution, and $\ell_y$ represents the cross-entropy loss function with convexity. **Our objective is to search for dynamic $w_v^d$ that minimizes the upper bound of $L_y$ as much as possible, and to prove that it is always superior to the static fusion weight $w_v^s = 1/V$.**

**Theorem 1** (Generalization Bound of Decision Fusion in Multi-View Concept-based Models)

Given a training set $\mathcal{D}_{\text{train}} = \left\{ \left( \mathbf{x}_i^{(1:V)}, \mathbf{c}_i^{(1:V)}, \mathbf{y}_i \right) \right\}_{i=1}^{N}$, we derive the generalization error bound of Multi-View Concept-based Models using Rademacher complexity (Bartlett & Mendelson, 2002), and for $1 > \delta > 0$, with probability at least $1 - \delta$, it holds that

$$
L_y \leq \underbrace{\sum_{v=1}^{V} \mathbb{E}[w_v]\hat{L}_y^{(v)} + \sum_{v=1}^{V} \mathbb{E}[w_v]L_g^{(v)}\hat{L}_c^{(v)}}_{\text{Term-L (average empirical loss of prediction and concept)}} + \underbrace{\sum_{v=1}^{V} \mathbb{E}[w_v]\mathfrak{R}_N(\mathcal{G}) + \sum_{v=1}^{V} \mathbb{E}[w_v]L_g^{(v)}\mathfrak{R}_N(\mathcal{C})}_{\text{Term-C (average complexity of prediction and concept)}}
$$

$$
+ \underbrace{\sum_{v=1}^{V} \text{Cov}\left( w_v, \ \ell_y\left( G(\mathbf{c}^{(v)}), \mathbf{y} \right) \right) + \sum_{v=1}^{V} L_g^{(v)} \text{Cov}\left(w_v, \left\| \hat{\mathbf{c}}^{(v)} - \mathbf{c}^{(v)} \right\|_1 \right)}_{\text{Term-Cov (covariance between fusion weights and losses)}} + \underbrace{2P\sqrt{\frac{\ln(V/\delta)}{N}}}_{\text{concentration term}}, \quad (12)
$$

where $\hat{L}_y^{(v)}$ and $\hat{L}_c^{(v)}$ denote the empirical prediction error (evaluated under true concepts) and the empirical concept error, respectively, $\mathfrak{R}_N(\mathcal{G})$ and $\mathfrak{R}_N(\mathcal{C})$ denote the Rademacher complexities estimated with $N$ samples, $L_g^{(v)} > 0$ is the Lipschitz constant of $G$ with respect to its concept input (i.e., the sensitivity bound of the prediction loss with respect to the concept), $\text{Cov}(\cdot, \cdot)$ denotes the covariance, and $P > 0$ is an absolute constant determined by the boundedness of the loss. In particular, when $w_v = w_v^s = 1/V$, the Term-Cov becomes 0.

First, since $\hat{L}_y^{(v)}$, $\hat{L}_c^{(v)}$, $L_g^{(v)}$, $\mathfrak{R}_N(\mathcal{G}_v)$, and $\mathfrak{R}_N(\mathcal{H}_v)$ are trained within the same loss function class and are independent of $w_v$, for $0 < \delta < 1$, with probability at least $1 - \delta$, to ensure that the generalization bound $L_y$ under $w_v^d$ is smaller than that under $w_v^s$, it is required that:

$$
\underbrace{\mathbb{E}[w_v^d] \stackrel{.}{=} w_v^s}_{\text{always holds}}, \quad \text{Cov}\left( w_v, \ \ell_y\left( G(\mathbf{c}^{(v)}), \mathbf{y} \right) \right) \leq 0, \quad \text{Cov}\left(w_v, \left\| \hat{\mathbf{c}}^{(v)} - \mathbf{c}^{(v)} \right\|_1 \right) \leq 0. \quad (13)
$$

Although $\ell_y\left( G(\mathbf{c}^{(v)}), \mathbf{y} \right)$ denotes the prediction loss obtained from the true concepts, $\ell_y\left( G(\mathbf{c}^{(v)}), \mathbf{y} \right)$ and $\ell_y\left( G(\hat{\mathbf{c}}^{(v)}), \mathbf{y} \right)$ are positively correlated, since a smaller deviation between $\hat{\mathbf{c}}^{(v)}$ and $c^{(v)}$ leads to closer prediction behavior of $G(\hat{\mathbf{c}}^{(v)})$ and $G(\mathbf{c}^{(v)})$, which in turn results in similar values of the two losses. In addition, the concept loss (L1loss) is required to be negatively correlated $w_v$. Thus, we present the following corollary:

**Corollary 1** When fusion weight $w_v = w_v^d$ is negatively correlated with both the prediction loss and the concept loss of the view, the generalization bound of multi-view decision fusion can be reduced.

Inspired by (Zhang et al., 2023), the concept loss and grading prediction loss are observed to be positively correlated with uncertainty, and together with Corollary 1, we propose the dual uncertainty-driven multi-view fusion decision.

### 2.3.2 DUAL UNCERTAINTY-AWARE MODULE

For each view, we quantify concept- and grading-level uncertainty under the evidential framework of Subjective Logic, which parameterizes belief masses via a Dirichlet distribution (Shafer, 1976; Han et al., 2022). For concept-level modeling, we treat each concept as a binary classification. The evidence vector $\mathbf{e}_{v,c_j} = [e_{v,c_j}^+, e_{v,c_j}^-] = softplus(\hat{\mathbf{c}}^{(v)})$ yields Dirichlet parameters $\alpha_{v,c_j}^+ = e_{v,c_j}^+ + 1$ and $\alpha_{v,c_j}^- = e_{v,c_j}^- + 1$. The belief masses and uncertainty for concept $j$ in view $v$ are:

$$
b_{v,c_j}^+ = \frac{\alpha_{v,c_j}^+ - 1}{S_{v,c_j}}, \quad b_{v,c_j}^- = \frac{\alpha_{v,c_j}^- - 1}{S_{v,c_j}}, \quad \psi_{v,c_j}^{\text{con}} = \frac{2}{S_{v,c_j}}, \quad (14)
$$

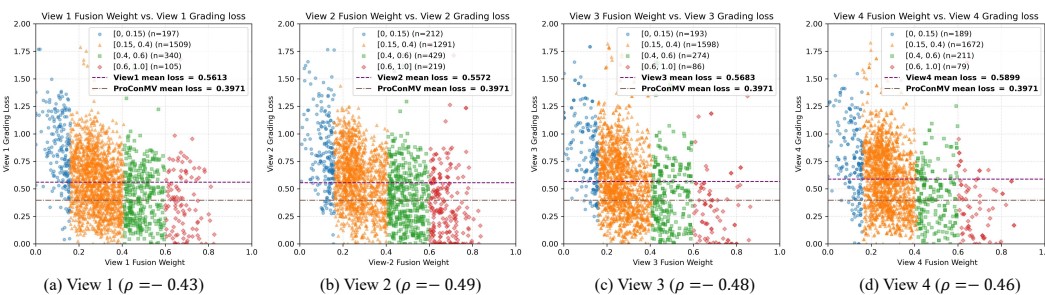

(a) View 1 ($\rho = -0.43$)  (b) View 2 ($\rho = -0.49$)  (c) View 3 ($\rho = -0.48$)  (d) View 4 ($\rho = -0.46$)

Figure 3: Scatter plots of per-view grading loss versus dual-uncertainty fusion weights on the MFIDDR test set, where $\rho$ denotes their Pearson correlation coefficient (Benesty et al., 2009).

where $S_{v,c_j} = \alpha^+_{v,c_j} + \alpha^-_{v,c_j}$. The overall concept uncertainty for view $v$ is averaged over all $m$ concepts: $\Psi^{\text{con}}_v = \frac{1}{m} \sum_{j=1}^{m} \psi^{\text{con}}_{v,c_j}$. For grading-level modeling with $K$ classes, the evidence vector $\mathbf{e}^{\text{gr}}_v = [e^{(1)}_v, \dots, e^{(K)}_v] = \text{softplus}(\hat{\mathbf{y}}^{(v)})$ gives $\alpha^{(i)}_v = e^{(i)}_v + 1$. The belief mass for grade $i$ and the grading uncertainty are:

$$b^{(i)}_v = \frac{\alpha^{(i)}_v - 1}{S^{\text{gr}}_v}, \quad \Psi^{\text{gr}}_v = \frac{K}{S^{\text{gr}}_v}, \tag{15}$$

with total strength $S^{\text{gr}}_v = \sum_{i=1}^{K} \alpha^{(i)}_v$, satisfying $\sum_{i=1}^{K} b^{(i)}_v + \Psi^{\text{gr}}_v = 1$.

### 2.3.3 MULTI-VIEW DECISION-MAKING UNDER DUAL UNCERTAINTIES

To construct a fully interpretable multi-view fundus decision model, our method exploits the uncertainties $\Psi^{\text{con}}_v$ and $\Psi^{\text{gr}}_v$ to assess view reliability, which in turn guides the dynamic fusion of outputs across views. In particular, the final grading decision $\hat{\mathbf{y}}$ is obtained by summing the view-specific outputs $\hat{\mathbf{y}}^{(v)}$, each weighted by a reliability score that combines concept- and grading-level certainties, $(1 - \Psi^{\text{con}}_v)$ and $(1 - \Psi^{\text{gr}}_v)$, with a learnable parameter $W_c = \frac{e^{-\eta}}{1 + e^{-\eta}} > 0$ controlling their trade-off:

$$\hat{\mathbf{y}} = \sum_{i=1}^{V} [W_c(1 - \Psi^{\text{con}}_v) + (1 - W_c)(1 - \Psi^{\text{gr}}_v)] \odot \hat{\mathbf{y}}^v. \tag{16}$$

This inverse dual-uncertainty design dynamically reduces the contribution of views with high uncertainty and low interpretability. As shown in Fig. 3, scatter plots and Pearson correlation coefficient $\rho$ intuitively demonstrate the correlation between per-view grading loss and dual-uncertainty fusion weights. This indicates that dual uncertainty can reduce the interference of unreliable perspectives on the final diagnosis and provide interpretable evidence for multi-view diagnosis.

### 2.4 LOSS FUNCTION

The training objective combines concept-level supervision for each view and the overall grading supervision. Specifically, we minimize

$$\mathcal{L} = -\sum_{j=1}^{n} (1 - \hat{\mathbf{y}}_j)^\gamma \, \mathbf{y}_j \log \hat{\mathbf{y}}_j - \frac{\alpha}{V} \sum_{i=1}^{V} \sum_{j=1}^{N} \left[ \mathbf{c}^{(v)}_j \log \hat{\mathbf{c}}^{(v)}_j + \left(1 - \mathbf{c}^{(v)}_j\right) \log \left(1 - \hat{\mathbf{c}}^{(v)}_j\right) \right]. \tag{17}$$

Here, the first term corresponds to the *Focal Loss* (with focusing parameter $\gamma$) for class-imbalanced DR grading, and the second term corresponds to the *Binary Cross-Entropy Loss* for concept prediction, with $\alpha$ balancing the two. By jointly optimizing both terms, the model is encouraged to learn faithful concept representations while simultaneously improving the final grading performance. Detailed hyperparameter experiments are presented in Fig. 8.

### 2.5 MULTI-VIEW TEST TIME INVENTION

Building upon our reasoning chain and dual-uncertainty decision paradigm, we propose a multi-view intervention mechanism that enables physicians to intervene on either a single view or a specific

Table 1: Comparison of Accuracy, Specificity, Kappa, and Macro F1-score on MFIDDR and DRTiD (Unit: %), and inference time (Unit: ms) for different models on DR grading. The best results are highlighted in bold, and "(MV)" means transforming into a multi-view method.

| Method | Venue | Backbone | MFIDDR (four views) | | | | DRTiD (two views) | | | | Infer. |
| | | | Acc.↑ | Spe.↑ | Kap.↑ | F1↑ | Acc.↑ | Spe.↑ | Kap.↑ | F1↑ | Time↓ |
|---|---|---|---|---|---|---|---|---|---|---|---|
| Non-interpretable Multi-View DR Diagnosis Methods | | | | | | | | | | | |
| CrossFit | BIBM'22 | Resnet-50 | – | – | – | – | 72.73 | 86.63 | 57.60 | 70.53 | – |
| ETMC | TPAMI'22 | Resnet-50 | 81.54 | 83.44 | 64.76 | 79.74 | 65.48 | 78.14 | 44.79 | 61.35 | **6.61** |
| MVCINN | AAAI'23 | Resnet-50+ViT | 80.10 | 83.32 | 62.45 | 78.86 | 68.18 | 85.78 | 51.39 | 66.83 | 31.31 |
| CVSRA-ViT | PR'25 | VGG+ViT | 82.61 | 86.77 | 68.57 | 81.94 | 70.62 | 88.91 | 55.74 | 69.97 | 71.53 |
| SMVDR | AAAI'25 | Mamba | 84.01 | 91.30 | 71.36 | 83.69 | 74.52 | 92.29 | 61.38 | 72.86 | 65.71 |
| WMIMVDR | ICME'25 | Resnet-50+ViT | 84.15 | 89.95 | 71.16 | 83.59 | 73.23 | 90.58 | 58.87 | 70.62 | 25.44 |
| Interpretable Multi-View DR Diagnosis Methods | | | | | | | | | | | |
| Multi-Task | TKDE'21 | Resnet-50 | 83.73 | 89.06 | 70.24 | 83.12 | 72.79 | 89.32 | 56.98 | 70.12 | 8.24 |
| MVCBM | ICML'22 | Resnet-50 | 83.22 | 88.22 | 69.12 | 82.43 | 71.54 | 85.02 | 57.89 | 68.50 | 19.67 |
| CEM (MV) | NIPS'22 | Resnet-50 | 84.12 | 88.77 | 70.83 | 83.45 | 74.55 | 91.67 | 61.42 | 72.06 | 21.25 |
| PCBM (MV) | ICML'23 | Resnet-50 | 83.52 | 91.05 | 70.35 | 83.29 | 74.73 | 90.26 | 60.68 | 71.99 | 17.56 |
| SSMVCBM | MIA'24 | Resnet-50 | 82.75 | 85.81 | 67.55 | 81.51 | 73.98 | 91.74 | 60.01 | 70.81 | 20.71 |
| CLAT (MV) | TMI'25 | ViT | 82.89 | 86.66 | 68.15 | 81.88 | 74.55 | 91.33 | 61.03 | 72.77 | 33.02 |
| ProConMV (Ours) | – | **Hilbert-RWKV** | **86.75** | **92.79** | **76.05** | **86.35** | **76.77** | **93.77** | **64.47** | **74.64** | 8.77 |

concept. Our method not only retains the ability of single-view CBMs to intervene on concepts to influence single-view decisions, but also leverages dual uncertainties at both the concept and grading levels to increase the contribution of the corresponding view to the overall decision. Specifically, taking view $i$ as an example, if an ophthalmologist corrects the result $\hat{\mathbf{c}}^{(v)}$ of lesion concepts in this view, the DR grading can first be re-inferred and updated as $\hat{\mathbf{y}}^{(v)}$, after which the dual uncertainties of the view are updated accordingly, thereby influencing the fused diagnostic outcome $\hat{\mathbf{y}}'$.

Table 2: Comparison of AUPR, AUC, Accuracy, Macro F1-score, Ranking Loss, and Hamming Loss on MFIDDR and DRTiD for different models on lesion concept classification. The best results are highlighted in bold, and "(MV)" means transforming into a multi-view method. (Unit: %)

| Method | MFIDDR | | | | | DRTiD | | | | |
| | AUPR↑ | ACC↑ | F1↑ | RL↓ | HL↓ | AUPR↑ | ACC↑ | F1↑ | RL↓ | HL↓ |
|---|---|---|---|---|---|---|---|---|---|---|
| Multi-Task | 54.69 | 93.87 | 51.69 | 3.73 | 5.54 | 47.32 | 87.31 | 43.90 | 7.88 | 12.69 |
| MVCBM | 61.56 | 94.22 | 59.10 | 3.21 | 5.36 | 48.50 | 88.82 | 41.88 | 5.56 | 11.18 |
| CEM (MV) | 65.47 | 94.74 | 60.42 | 2.86 | 5.21 | 48.50 | 89.95 | 44.63 | 6.12 | 11.05 |
| PCBM (MV) | 68.12 | 94.85 | 66.08 | 1.91 | 4.96 | 52.59 | 90.46 | 47.24 | 4.52 | 9.54 |
| SSMVCBM | 66.25 | 94.42 | 63.34 | 2.17 | 5.02 | 53.52 | 90.35 | 47.15 | 4.25 | 9.65 |
| CLAT (MV) | 63.89 | 94.63 | 59.15 | 2.98 | 5.45 | 51.83 | 89.97 | 46.82 | 4.71 | 10.02 |
| ProConMV (Ours) | **72.26** | **95.42** | **68.43** | **1.45** | **4.47** | **55.86** | **90.83** | **48.00** | **3.42** | **9.17** |

## 3 EXPERIMENTS

### 3.1 EXPERIMENTAL SETTINGS

**Datasets.** We evaluate our method on the two publicly available multi-view DR grading datasets, MFIDDR (Luo et al., 2023) and DRTiD (Hou et al., 2022). MFIDDR contains 34,452 images from 4,344 patients, annotated with five DR grades across four standard views (macula-centered, optic disc–centered, and superior/inferior tangent to the optic disc). DRTiD consists of 3,100 paired macula- and optic disc–centered images from 1,550 eyes. To enable concept-based reasoning, ophthalmologists annotate six lesion concepts in the fundus images of each dataset, which serve as concept prediction labels: hard exudates (EX), hemorrhages (HE), microaneurysms (MA), soft exudates (SE), vitreous hemorrhage (VH), and vitreous opacity (VO). For fair comparison, we follow the official data split protocols provided by each dataset, respectively. Detailed statistics of the dataset distributions are summarized in Section A.3. And lesion masks are generated by the HACDR-Net (Xu et al., 2024) pre-trained on DDR Li et al. (2019) dataset.

**Evaluation Metric and Compared Methods.** In this section, we evaluate multi-view DR diagnosis on two tasks: multi-view DR grading and lesion concept classification. Grading is assessed using accuracy (Acc.), precision (Prec.), sensitivity (Sens.), specificity (Spe.), kappa, macro-F1, and AUC (Trevethan, 2017), while the lesion concept classification uses AUPR, AUC, Acc., macro-F1, ranking loss (RL), and Hamming loss (HL). We also report inference time. Baselines are divided into two categories: (i) non-interpretable multi-view DR diagnosis methods, including CrossFit (Hou et al., 2022), ETMC (Han et al., 2022), MVCINN (Luo et al., 2023), Retfound (Zhou et al., 2023), CVSRA-ViT (Lin et al., 2025), SMVDR (Luo et al., 2025), and WMIMVDR (Hu et al., 2025), and (ii) interpretable multi-view DR diagnosis methods, including Multi-Task (Zhang & Yang, 2022), MVCBM (Klimiene et al., 2022), CEM (Espinosa Zarlenga et al., 2022), PCBM-h (Yuksekgonul et al., 2023), SSMVCBM (Marcinkevičs et al., 2024), and CLAT (Wen et al., 2024).

**Implementation Details.** All experiments are implemented with PyTorch and conducted on an NVIDIA RTX 4090 GPU. We resize the images and labels to a resolution of $256 \times 256$. The batch size and number of epochs are set to 8 and 100, respectively. The Adam optimizer is used with an initial learning rate of $10^{-5}$, which is dynamically adjusted by a cosine annealing scheduler. We select the model achieving the best grading performance on the validation set as the final model, which is then used for subsequent testing and analysis.

## 3.2 EXPERIMENTAL ANALYSIS

**Comparison with Advanced Methods.** We compare our method with 12 state-of-the-art multi-view methods on two datasets. As shown in Table 1, our ProConMV achieves the best performance in multi-view DR grading on both datasets. Specifically, ProConMV improves accuracy by 2.6% on the four-view dataset and by 2.04% on the two-view dataset, with the highest Kappa improvement of 4.69 on MFIDDR. As presented in Table 2, our method also achieves the best results in lesion concept classification on both datasets. The AUPR is improved by 4.14% and 2.34% on MFIDDR and DRTiD, respectively, while RL and HL are significantly reduced. Benefiting from the linear complexity of Hilbert-RWKV, our method also ranks among the top in single-image inference time. Overall, our method achieves the highest accuracy in both grading and concept prediction for the diagnostic task, while also delivering superior inference efficiency compared to most existing interpretable and non-interpretable multi-view diagnostic methods.

**Analysis of Test-time Intervention Capability.** As a concept-based model, our approach enables interventions from both the view and lesion perspectives. Specifically, ophthalmologists can revise the diagnosis of a particular view, a specific lesion type, or the same lesion type across multiple views, and such interventions directly refine the final DR grading outcome. As shown in Fig. 4, the overall grading accuracy increases proportionally with the number of intervened views and lesion concepts. When all erroneous concepts are corrected, the grading accuracy reaches 91.56%, yielding a 4.81% improvement over the non-intervention setting. This verifies the effectiveness of the proposed intervention method.

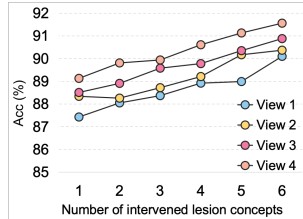

Figure 4: Performance evaluation of the multi-view test-time intervention.

## 3.3 ABLATION STUDY

**Ablation Results of Hilbert-RWKV.** We first compare Hilbert-RWKV with existing state-of-the-art backbones, as shown in Table 4. Compared to ResNet-50 (He et al., 2016), ViT-Big (Dosovitskiy et al., 2021), VMamba-Tiny (Liu et al., 2024b), Swin v1-S Liu et al. (2021) and Swin v2-S Liu et al. (2022b), our method achieves consistent improvements in DR grading and lesion classification with lower parameter counts and shorter inference times. For example, our method surpasses ResNet-50 by 1.03% Acc. and 2.27% F1 in DR grading, and by 2.80% AUPR and 2.80% F1 in lesion classification, while requiring significantly fewer parameters (6.70M vs. 25.26M) and less inference time (8.77ms vs. 10.16ms). Subsequently, as presented in Table 5, we evaluate different scanning strategies for the RWKV architecture. Compared with Hilbert scanning, sweep, zigzag, and unidirectional scanning lead to drops of 0.78% and 2.18% in F1 scores for DR grading and lesion classification, respectively. These results substantiate the superiority of the Hilbert-RWKV design in both DR diagnostic reasoning and computational efficiency.

Table 3: Ablation studies of key modules on MFIDDR. '×' and '✓' denote the absence and presence of each module. The first row corresponds to the baseline, MVCBM. (Unit: %)

| Mask | Text | Hilbert-RWKV | VT-RWKV | DU-MVFD | Grading Acc | Grading Kappa | Concept Pred. AUPR | Concept Pred. F1 |
|---|---|---|---|---|---|---|---|---|
| ✓ | ✓ | × | × | × | 83.22 | 69.12 | 61.56 | 59.10 |
| ✓ | ✓ | ✓ | × | × | 85.72 | 74.31 | 62.90 | 63.87 |
| ✓ | ✓ | × | ✓ | × | 85.45 | 74.29 | 64.72 | 62.61 |
| ✓ | ✓ | × | × | ✓ | 84.15 | 71.57 | 60.87 | 58.92 |
| ✓ | ✓ | ✓ | ✓ | × | 86.23 | 75.23 | 70.69 | 67.89 |
| ✓ | ✓ | ✓ | × | ✓ | 85.68 | 74.43 | 70.12 | 64.39 |
| ✓ | ✓ | × | ✓ | ✓ | 85.12 | 72.68 | 67.71 | 66.12 |
| ✓ | × | ✓ | ✓ | ✓ | 85.68 | 74.43 | 70.12 | 64.39 |
| × | ✓ | ✓ | ✓ | ✓ | 85.21 | 73.98 | 68.59 | 66.26 |
| ✓ | ✓ | ✓ | ✓ | ✓ | **86.75** | **76.05** | **72.26** | **68.43** |

Table 4: Comparison of backbones. (Unit: %)

| Backbone | Grading Acc. | Grading F1 | Concept Pred. AUPR | Concept Pred. F1 | Params (M) | Infer. (ms) |
|---|---|---|---|---|---|---|
| VGG-16 | 85.63 | 85.26 | 67.69 | 66.88 | 15.29 | 6.93 |
| ResNet-50 | 85.72 | 84.08 | 67.71 | 66.12 | 25.26 | 10.16 |
| ViT-B | 85.82 | 85.44 | 55.01 | 52.81 | 86.61 | 12.09 |
| Swin v1-S | 86.09 | 86.03 | 66.65 | 60.09 | 49.56 | 18.65 |
| Swin v2-S | 86.33 | 85.76 | 61.47 | 57.52 | 37.93 | 26.95 |
| VMamba | 83.03 | 82.29 | 59.43 | 52.15 | 14.60 | 9.27 |
| Ours | **86.75** | **86.35** | **72.26** | **68.92** | **6.70** | 8.77 |

Table 5: Comparison of scanning strategies on MFIDDR. (Unit: %)

| Strategy | Grading Acc. | Grading F1 | Concept Pred. F1 | Concept Pred. AUPR |
|---|---|---|---|---|
| Sweep | 86.29 | 85.94 | 65.24 | 69.61 |
| Horizontal | 86.24 | 86.05 | 64.93 | 70.97 |
| Vertical | 86.10 | 85.96 | 64.75 | 70.46 |
| Zigzag | 86.24 | 85.70 | 63.89 | 67.60 |
| Ours | **86.75** | **86.35** | **68.92** | **72.26** |

**Ablation Results of VT-RWKV and multi-view fusion method.** As shown in Fig. 5(a), we compare different interaction strategies for VT-RWKV, where Cat denotes channel-wise fusion, Attn applies cross-attention, and DyF (Xue & Marculescu, 2023) uses dynamic multimodal fusion. VT-RWKV consistently outperforms these designs, achieving improvements of +0.70% ACC, +1.13% Kappa, +0.72% AUPR, and +1.61% F1, confirming the superiority of our image-text concept interaction. For multi-view fusion (Fig. 5(b)), our dual-uncertainty module achieves the best performance, outperforming Baseline (Concat), Late Fusion (Average add), TMC, and Moe (Cao et al., 2023) with gains of +0.86% Acc, +1.97% Kappa, +2.57% AUPR, and +1.77% F1. This demonstrates the effectiveness of uncertainty modeling for multi-view decision fusion.

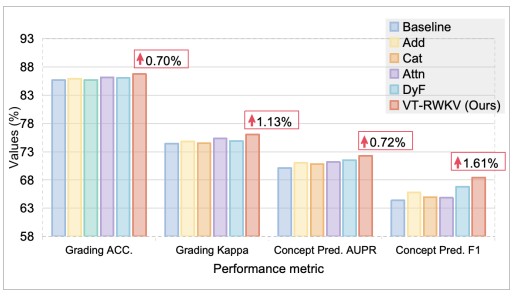

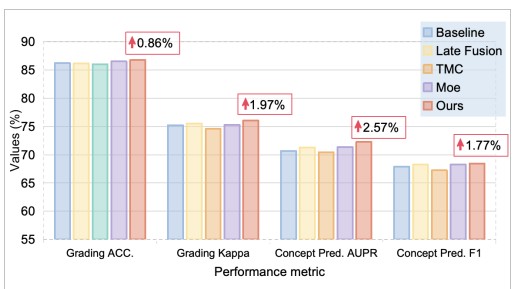

(a) Comparison of experimental results of different image-text concept interaction methods.

(b) Comparison of experimental results of different ways of multi-view fusion decision.

Figure 5: Ablation studies on VT-RWKV and DU-MVFD on the MFIDDR dataset.

## 4 CONCLUSION

This paper proposes a full-process interpretable model, ProConMV. It achieves deep extraction and fusion of multi-source features, introduces lesion concepts to construct a causal reasoning chain, and incorporates real-time physician intervention. Moreover, the proposed multi-view decision-making approach theoretically reduces generalization error and achieves traceability through a dual uncertainty module. The evaluation results show that it achieves state-of-the-art performance and high clinical credibility in DR grading. Future research will focus on weakly-supervised learning with sparse data and conduct large-scale studies involving physician users.

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

# A APPENDIX

## A.1 RELATED WORK

### A.1.1 DNN-BASED METHODS FOR MULTI-VIEW DIABETIC RETINOPATHY DIAGNOSIS

Recently, multi-view approaches for DR diagnosis have attracted increasing attention. Luo et al. (Luo et al., 2023) first proposed MVCINN, a multi-view DR diagnosis network that integrates CNNs and Transformers. Subsequently, several works (Luo et al., 2024; Lin et al., 2025; Hu et al., 2025) have leveraged visual cues, such as vessel and DR lesion masks derived from segmentation models, to improve diagnostic accuracy. Others (Hou et al., 2022; Luo et al., 2025) focused on inter-view information exchange and backbone design to further strengthen diagnostic representations. Nevertheless, interpretability for this task has not been adequately explored, particularly in terms of textual explanations and transparent diagnostic workflows, which are of great significance in clinical medical diagnosis.

### A.1.2 INTERPRETABLE MACHINE LEARNING MODELS IN COMPUTER VISION

Interpretability methods have achieved remarkable success in computer vision, which enhances human understanding of model predictions. Early studies on interpretability mainly focused on post-hoc explanations of black-box models, such as Shapley (Roth, 1988; Chen et al., 2022), Grad-CAM (Selvaraju et al., 2017; Chattopadhay et al., 2018), and Prototypes (Seo et al., 2023). However, these methods lack human-comprehensible reasoning processes and are therefore fundamentally unable to provide reasonable explanations for downstream vision applications. To this end, Koh et al. (Koh et al., 2020) proposed the Concept Bottleneck Model (CBM), an interpretable framework that first predicts visual concepts and then uses them to generate the final prediction. There is a diverse set of CBM variants (Espinosa Zarlenga et al., 2022; Zhang et al., 2024; Wen et al., 2024; Ciravegna et al., 2022; Sun et al., 2025), each tackling the problem from a different perspective. To the best of our knowledge, SSMVCBM (Marcinkevičs et al., 2024) is the only work on interpretable multi-view classification. In contrast to prior concept-based studies, grounded in the perspective of interpretable multi-view vision task, our work 1) introduces a multimodal RWKV module to enhance concept representations, and 2) proposes a dual–uncertainty–aware fusion strategy that explicitly accounts for both concept and outcome uncertainties in multi-view decision-making.

### A.1.3 RECEPTANCE WEIGHTED KEY VALUE

Receptance Weighted Key Value (RWKV) (Peng et al., 2023) is a neural network architecture that combines the parallel training ability of Transformers with the efficient recurrence of RNNs, characterized by its linear complexity and effectiveness in modeling long sequences. Recently, RWKV has gained renewed attention in vision tasks, as its core WKV attention mechanism has demonstrated superior performance compared to self-attention (Dosovitskiy et al., 2021) in some vision domains. Duan et al. proposed Vision-RWKV (Duan et al., 2025), first introducing a bidirectional WKV attention mechanism and a quad-directional token shift method to adapt RWKV for image classification tasks. Building upon RWKV and Vision-RWKV, a series of variants have been introduced for diverse vision-related tasks, including RWKV-SAM (Yuan et al., 2024) for segmentation, RWKV-CLIP (Gu et al., 2024) for vision-language representation learning, and Point-RWKV (He et al., 2025) for 3D point cloud learning. However, these works overlook the problem of spatial locality loss introduced by token serialization in image modeling.

## A.2 PROOF OF THEOREM 1

According to the definitions and settings mentioned above, based on the convexity of the prediction loss $\ell_y^{(v)}(\cdot, \cdot)$ and the normalization property of $w_v$, we can derive:

$$\ell_y\left(\sum_{v=1}^{V} w_v\, G(\hat{\mathbf{c}}^{(v)}),\, \mathbf{y}\right) \leq \sum_{v=1}^{V} w_v \ell_y(G(\hat{\mathbf{c}}^{(v)}),\, \mathbf{y}) \tag{18}$$

Using the Lipschitz constraint, we decompose $\ell_y\big(G(\hat{\mathbf{c}}^{(v)}), \mathbf{y}\big)$ as follows:

$$\ell_y\big(G(\hat{\mathbf{c}}^{(v)}), \mathbf{y}\big) \;\leq\; \ell_y\big(G(\mathbf{c}^{(v)}), \mathbf{y}\big) + L_g^{(v)}\big\|\hat{\mathbf{c}}^{(v)} - \mathbf{c}^{(v)}\big\|_1 \;\leq\; \ell_y\big(G(\mathbf{c}^{(v)}), \mathbf{y}\big) + L_g^{(v)}\,\ell_c^{(v)}. \tag{19}$$

By combining Equation (18) and (19), the upper bound of $L_y$ can be rewritten as:

$$L_y \leq \sum_{v=1}^{V} \mathbb{E}\Big[ w_v\, \ell_y\big(G(\hat{\mathbf{c}}^{(v)}), y\big)\Big] \leq \sum_{v} \mathbb{E}\Big[ w_v\, \ell_y\big(G(\mathbf{c}^{(v)}), \mathbf{y}\big)\Big] \;+\; \sum_{v=1}^{V} L_g^{(v)}\, \mathbb{E}\Big[ w_v\, \ell_c^{(v)}\Big]. \quad (20)$$

According to the property of expectation, for any random variables $A$ and $B$, $\mathbb{E}[AB] = \mathbb{E}[A]\,\mathbb{E}[B] + \text{Cov}(A, B)$.

$$L_y \leq \sum_{v=1}^{V} \big(\mathbb{E}[w_v]\,\mathbb{E}[\ell_y(G(\mathbf{c}^{(v)}), \mathbf{y})]\big) + \sum_{v=1}^{V} \big(\mathbb{E}[w_v]\, L_g^{(v)}\, \mathbb{E}[\ell_c^{(v)}]\big)$$
$$+ \sum_{v=1}^{V} \text{Cov}(w_v, \ell_y(G(\mathbf{c}^{(v)}), \mathbf{y})) + \sum_{v=1}^{V} L_g^{(v)}\, \text{Cov}(w_v, \ell_c^{(v)}). \quad (21)$$

To simplify Equation (21), we take $\mathbb{E}[\ell_y\big(G(\mathbf{c}^{(v)}), \mathbf{y}\big)]$ as an example and invoke Rademacher complexity theory, which establishes that with a confidence level $1 - \delta$, where $0 < \delta < 1$, the following holds:

$$\mathbb{E}[\ell_y(G(\mathbf{c}^{(v)}), \mathbf{y})] \;\leq\; \hat{L}_y^{(v)} \;+\; \mathfrak{R}_N(\mathcal{G}) \;+\; P\sqrt{\tfrac{\ln(V/\delta)}{N}}. \quad (22)$$

Where $\hat{L}_y^{(v)}$ denotes the empirical prediction error under correct concepts. Similarly, it can be derived that: $\mathbb{E}[\ell_c^v] \;\leq\; \hat{L}_c^{(v)} \;+\; \mathfrak{R}_N(\mathcal{H}_v) \;+\; P\sqrt{\tfrac{\ln(V/\delta)}{N}}$. In summary, we can obtain the final theorem:

$$L_y \leq \underbrace{\sum_{v=1}^{V} \mathbb{E}[w_v]\hat{L}_y^{(v)} + \sum_{v=1}^{V} \mathbb{E}[w_v] L_g^{(v)}\hat{L}_c^{(v)}}_{\text{Term-L (average empirical loss of prediction and concept)}} + \underbrace{\sum_{v=1}^{V} \mathbb{E}[w_v]\mathfrak{R}_N(\mathcal{G}) + \sum_{v=1}^{V} \mathbb{E}[w_v] L_g^{(v)}\mathfrak{R}_N(\mathcal{C})}_{\text{Term-C (average complexity of prediction and concept)}}$$

$$+ \underbrace{\sum_{v=1}^{V} \text{Cov}(w_v, \ell_y\big(G(\mathbf{c}^{(v)}), \mathbf{y}\big)) + \sum_{v=1}^{V} L_g^{(v)}\, \text{Cov}(w_v, \ell_c^{(v)})}_{\text{Term-Cov (covariance between fusion weights and losses)}} + \underbrace{2P\sqrt{\tfrac{\ln(V/\delta)}{N}}}_{\text{concentration term}}. \quad (23)$$

### A.3 STATISTICAL INFORMATION OF TWO DATASETS

In Fig. 6, we list the detailed information of two DR diagnosis datasets used in our experiments. The relationships represented by the Sankey diagram capture underlying DR diagnostic rules, which in turn validate the rationality and interpretability of our reasoning model.

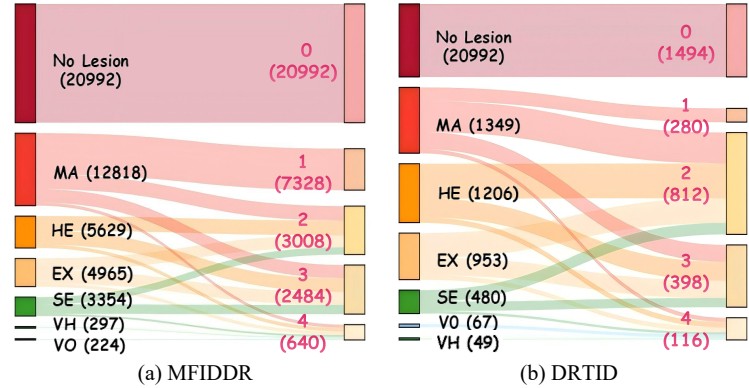

(a) MFIDDR             (b) DRTID

Figure 6: The data correlation and distribution of lesion concepts and DR grades in the two benchmarks. On the left are lesion concepts, and on the right are DR grades. The text indicates the class name, and the number in parentheses denotes the number of samples in each class.

Table 6: Comparison of Precision and Macro F1-score of different methods for DR 0-4 grades on MFIDDR. The best results are highlighted in bold, and "(MV)" means transforming into a multi-view method. (Unit: %)

| Method | Grade 0 | | Grade 1 | | Grade 2 | | Grade 3 | | Grade 4 | | Avg. | |
|---|---|---|---|---|---|---|---|---|---|---|---|---|
| | Prec.↑ | F1↑ | Prec.↑ | F1↑ | Prec.↑ | F1↑ | Prec.↑ | F1↑ | Prec.↑ | F1↑ | Prec.↑ | F1↑ |
| ETMC | 86.79 | 91.85 | 73.26 | 63.72 | **66.41** | 55.41 | 64.41 | 70.15 | 0.12 | 0.87 | 58.20 | 56.40 |
| MVCINN | 86.71 | 91.26 | 68.25 | 56.43 | 57.44 | 59.26 | 70.00 | 68.06 | 68.42 | **44.83** | 70.16 | 63.98 |
| Retfound | 80.11 | 87.47 | 50.20 | 35.92 | 54.41 | 46.39 | 65.79 | 66.67 | 90.00 | 36.73 | 68.10 | 54.64 |
| SMVDR | 93.48 | 93.52 | 71.15 | 71.70 | 60.00 | 60.33 | 69.41 | 74.21 | 99.99 | 30.43 | 78.81 | 66.04 |
| WMIMVDR | 92.26 | 93.49 | 71.02 | 71.41 | 63.98 | 59.88 | **71.87** | **74.68** | 87.50 | 29.79 | 77.33 | 65.85 |
| Multi-Task | 91.43 | 93.24 | 71.60 | 69.87 | 64.12 | 61.76 | 69.81 | 72.31 | 87.50 | 29.79 | 76.89 | 65.39 |
| MVCBM | 90.55 | 93.31 | 75.77 | 67.08 | 57.58 | 59.84 | 67.44 | 72.50 | 87.50 | 29.79 | 75.77 | 64.50 |
| CEM (MV) | 91.10 | 93.46 | 73.53 | 70.18 | 63.64 | 62.40 | 71.61 | 73.27 | 84.75 | 35.40 | 76.93 | 66.94 |
| PCBM-h (MV) | 93.44 | 93.94 | 71.72 | 71.03 | 52.82 | 56.82 | 71.13 | 69.66 | 94.44 | 35.42 | 76.71 | 65.37 |
| SSMVCBM | 88.66 | 92.78 | 73.63 | 66.09 | 64.23 | 55.00 | 66.48 | 72.78 | 99.99 | 30.43 | 78.60 | 63.42 |
| CLAT (MV) | 89.30 | 92.68 | 73.17 | 66.18 | 64.42 | 60.69 | 67.24 | 72.67 | 99.99 | 26.67 | 78.83 | 63.78 |
| ProConMV (Ours) | **94.61** | **95.35** | **77.85** | **78.63** | 63.19 | **63.01** | 71.14 | 71.38 | 88.89 | 33.00 | **79.14** | **68.27** |

## A.4 SUPPLEMENTARY COMPARATIVE EXPERIMENTAL RESULTS

To better evaluate our proposed method, we compare it against 12 existing methods on MFIDDR dataset. Notably, We adapt RETFound to multi-view fundus image data by designing a multi-channel network to extract features from multiple views, which are then concatenated and fed into the classifier. As shown in Fig. 6, our method achieves the best performance on Grade 0 and Grade 1, with improvements of 1.13%, 1.41%, 2.08%, and 6.93% over the second-best results in Grade 0 precision, Grade 0 F1-score, Grade 1 precision, and Grade 1 F1-score. Moreover, the proposed method achieves competitive results on Grade 2 and Grade 3. However, it does not attain the best performance on Grade 4, possibly due to sample imbalance. Overall, our method achieves improvements of 5.68% in average precision and 4.76% in average F1-score compared to the mean values of the other twelve methods. Additionally, the visualization of the inference process of our proposed ProConMV model is shown in Fig. 9.

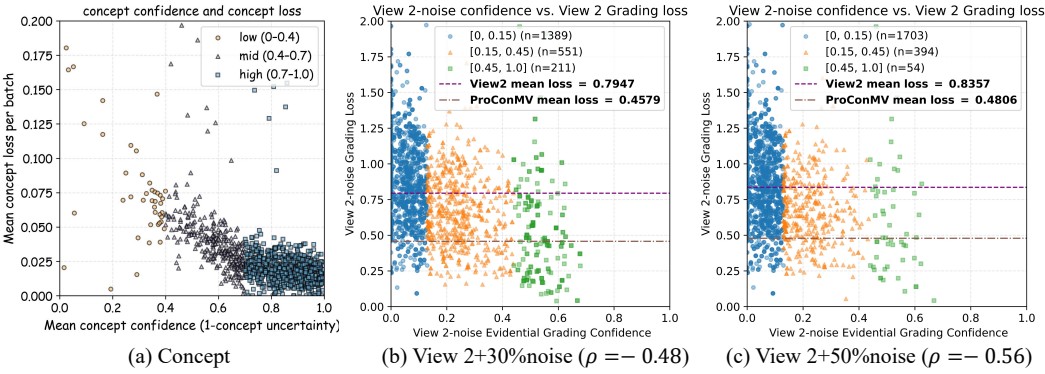

(a) Concept  (b) View 2+30%noise ($\rho = -0.48$)  (c) View 2+50%noise ($\rho = -0.56$)

Figure 7: (a) Scatter plot of concept confidence versus loss. (b) Scatter plot of confidence versus loss for View 2 after adding 30% Gaussian noise. (c) Scatter plot of confidence versus loss for View 2 after adding 50% Gaussian noise on MFIDDR test set. Here, $\rho$ denotes their Pearson correlation coefficient (Benesty et al., 2009).

## A.5 ANALYSIS OF DUAL-UNCERTAINTY-AWARE MULTI-VIEW FUSION

We provide scatter plots of the concept loss versus its confidence (1−concept uncertainty). Based on the Pearson correlation coefficient $\rho = -0.34$ and the trend observed in Fig. 7, we can see that they are negatively correlated. In addition, we evaluate the sensitivity of uncertainty and the rationality of our module design under noisy conditions. Under noise perturbations, the grading loss of View 2 increases, while its confidence drops significantly. The number of views with confidence in the

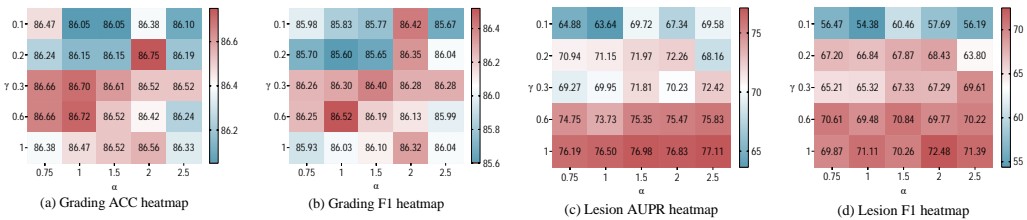

Figure 8: Results of the ablation study on the hyperparameters $\alpha$ and $\gamma$. The figure presents grading accuracy, grading F1-score, concept prediction accuracy, and concept prediction F1-score under different values of $\alpha$ and $\gamma$. The best results are indicated by the warmest colors.

range [0, 0.15] also increases significantly as the noise level rises from 30% to 50%, as shown in Fig. 3 and Fig. 7. Overall, these results demonstrate the reliability of our dual-uncertainty-aware multi-view fusion method.

## A.6 HYPERPARAMETER ANALYSIS

The hyperparameters $\alpha$ and $\gamma$ respectively control the weight of the concept prediction loss and the focusing parameter in the grading loss. To analyse the impact of these hyperparameters, we conduct a two-dimensional ablation study by systematically varying both $\alpha$ and $\gamma$. As shown in Fig. 8, when $\alpha$ increases, the contribution of the concept prediction loss becomes more prominent, resulting in higher concept prediction accuracy. However, a larger value of $\alpha$ reduces grading accuracy, which contradicts our goal of achieving optimal grading performance. Therefore, there is a trade-off between these two hyperparameters. Based on this analysis, we select $\alpha = 0.2$ and $\gamma = 2.0$ as the optimal hyperparameter settings.

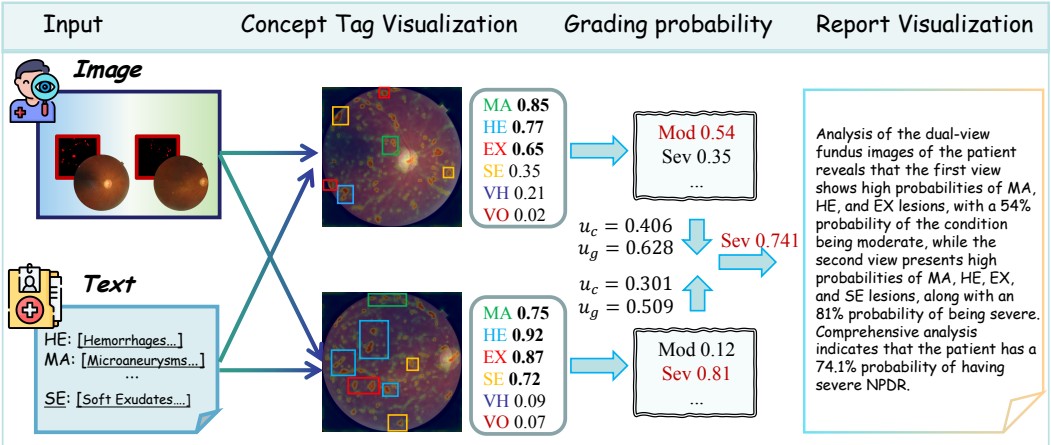

Figure 9: Visualization display of our ProConMV model's Inference Process.

## A.7 INTERPRETABILITY ANALYSIS

Fig. 9 illustrates that our model provides a fine-grained and clinically meaningful interpretation of its diagnostic workflow. For each fundus view, the model first localizes lesion regions and assigns concept tags (e.g., MA, HE, EX, SE, VH, VO) with calibrated probabilities, which are consistent with the accompanying textual descriptions of hemorrhages, microaneurysms, and soft exudates. In the first view, high probabilities of MA, HE, and EX lead to a moderate grading score, whereas in the second view, simultaneously elevated MA, HE, EX, and SE scores support a severe grade. The dual-uncertainty-aware fusion module then combines per-view concept and grading confidences to produce a final severe NPDR probability of 0.741, explicitly revealing how each view contributes to

the outcome. Finally, the system generates a natural-language report that summarizes the detected lesions, per-view severity, and overall diagnostic conclusion, allowing clinicians to verify the prediction step by step. Together, these visual and textual explanations show that the model's decision is driven by clinically interpretable lesion patterns rather than opaque global features.

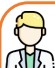
Briefly describe Hemorrhages' fundus appearance, key characteristics, and the DR stage(s) in which they most commonly occur (in 2–3 sentences each).

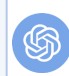
**Hemorrhages**
Retinal bleeding (dot, blot, flame-shaped) caused by ruptured microaneurysms or fragile capillaries. They appear as multiple deep red lesions in the inner retinal layers and increase in number and size as diabetic retinopathy progresses. According to standard grading rules, scattered hemorrhages are common in Moderate NPDR (Grade 2), while extensive hemorrhages in all four quadrants are a hallmark of Severe NPDR (Grade 3) under the '4-2-1' rule.

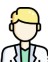
Briefly describe Hard Exudates' fundus appearance, key characteristics, and the DR stage(s) in which they most commonly occur (in 2–3 sentences each).

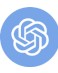
**Hard Exudates**
Yellow, well-defined lipid deposits resulting from chronic leakage of serum lipoproteins from incompetent retinal vessels or microaneurysms. They often cluster in circinate (ring-like) patterns around leaking points and can accumulate in the macula, threatening central vision. In grading, the presence of hard exudates is typical of Moderate NPDR and more advanced stages (Grade ≥2), and when located near the fovea, they are used to identify clinically significant macular edema in some grading schemes.

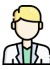
Briefly describe Microaneurysms' fundus appearance, key characteristics, and the DR stage(s) in which they most commonly occur (in 2–3 sentences each).

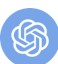
**Microaneurysms**
Small, round red dots produced by focal outpouching of capillary walls, representing the earliest clinically visible sign of diabetic retinopathy. They are usually located in the posterior pole and may be isolated or accompanied by minimal hemorrhages. In most grading systems, the presence of microaneurysms with few or no additional lesions corresponds to Mild NPDR (Grade 1), and increasing numbers of microaneurysms signal progression toward more severe stages.

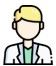
Briefly describe Soft Exudates' fundus appearance, key characteristics, and the DR stage(s) in which they most commonly occur (in 2–3 sentences each).

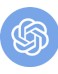
**Soft Exudates**
White, fluffy 'cotton-wool' spots arising from focal ischemia and axoplasmic flow stasis in the nerve fiber layer. They indicate localized retinal infarction and are often associated with widespread capillary non-perfusion. In diagnostic grading, the occurrence of multiple soft exudates together with extensive hemorrhages, venous beading, or intraretinal microvascular abnormalities (IRMA) supports a diagnosis of Severe NPDR (Grade 3) according to the '4-2-1' rule.

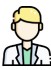
Briefly describe Vitreous Hemorrhages' fundus appearance, key characteristics, and the DR stage(s) in which they most commonly occur (in 2–3 sentences each).

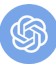
**Vitreous Hemorrhages**
Blood leakage into the vitreous cavity or preretinal space from fragile neovascular vessels that break through the internal limiting membrane. Clinically, they present as sudden vision loss or floaters and may partially or completely obscure the fundus. In grading systems, the presence of vitreous or preretinal hemorrhage due to neovascularization is a defining sign of proliferative diabetic retinopathy (PDR, Grade 4) and identifies high-risk eyes that require urgent treatment.

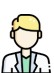
Briefly describe Vitreous Opacities' fundus appearance, key characteristics, and the DR stage(s) in which they most commonly occur (in 2–3 sentences each).

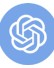
**Vitreous Opacities**
Cloudy or hazy areas within the vitreous caused by suspended blood, fibrous tissue, or fibrovascular membranes following recurrent hemorrhage and neovascular proliferation. These opacities can produce persistent floaters, reduced contrast sensitivity, and significant vision loss if they obscure the macula. In severity grading, marked vitreous opacities are characteristic of advanced PDR (Grade 4) and often indicate tractional changes that may progress to retinal detachment if not properly managed.

Figure 10: The generation of concept text.

## A.8    CONCEPT TEXT GENERATION

We obtain the concept descriptions of DR lesions through a six-round dialogue with GPT-4 (Achiam et al., 2023). In each round, the model extracts specific feature information about a lesion, ultimately generating an accurate medical text description. The generated text includes the lesion's typical appearance, key features, and its manifestations at different stages of DR, as shown in Fig. 10.

### A.9 LLM Usage Statement

In this paper, an LLM is used solely for text polishing. All research ideas, methods, experiments, and conclusions are conducted without the assistance of any LLMs.

