# OpenReview forum: "Provenance-Enabled Multi-View Diabetic Retinopathy Diagnosis Through Interpretable Process Mining"
_ICLR.cc/2026/Conference — ICLR 2026 Conference Withdrawn Submission_

### Official Review · Reviewer_eP2g · 2025-10-28

**Soundness:** 2
**Presentation:** 2
**Contribution:** 2
**Rating:** 4
**Confidence:** 5

**Summary:**

This paper presents a multi-view framework for diabetic retinopathy grading, incorporating concept-based reasoning, RWKV backbone, and an uncertainty-aware mechanism. The topic is highly relevant and the results are remarkable. However, there are several aspects that need further clarification and discussion to strengthen the overall narrative and contribution.

**Strengths:**

1. The topic discussed in the paper is important for AI-assisted medical analysis.
2. The propsoed method shows a promising performance to enhance the transparency of the diagnosis process.

**Weaknesses:**

###Major concerns
1. L130-131, 'We observe that existing Transformer-based multi-view methods are less effective at fine-grained local concept perception'. There appear to be no empirical results or evidence provided to support this claim. It is recommended to add relevant experimental comparison or references to prior work
2. The motivation behind introducing the "multi-directional Hilbert attention mechanism" is unclear. Is there any benefit to introducing Hilbert curve?
3. Why was TE3 chosen as the encoder instead of alternatives specifically designed for medical applications? Domain specific encoders are often tailored to extract relevant features, and it would be helpful to discuss why TE3 was preferred over others in this instance.
4. L236-238, "Through this fusion, the model obtains each view’s lesion concept embeddings that are aligned with both visual information and diagnostic knowledge, thereby enhancing the interpretability and predictive accuracy of the concepts." It is unclear why the interpretability could be enhanced by this fusion
5. How is a view determined to have "poor interpretability"? (L342) I don't think the uncertainty is equivalent with interpretability
6. The role and mechanism of the mask mentioned in various parts of the paper are vague. Where exactly is the mask applied, and how does it operate within the framework? Are those compared methods integrated with masks?
7. Is it possible to release the annotated concept labels? It is critical for reproducing the results.
8. The ablation results are substantial, but the impact of each module is not thoroughly discussed. e.g. DU-MVFD in Tab. 3. showed only a slight drop in concept prediction metrics compared to the baseline. Yet, significant performance gains were observed when combined with Hilbert-RWKV and VT-RWKV. It is recommended to have more discussion highlighting the effect of modules and the combination of them to help readers understand the mechanism
9. My main concern centers on interpretability, which appears somewhat contradictory to the paper's core contribution. While the multiview setup improves accuracy and it is intuitive, it actually introduces negative implications for interpretability since decision-making based on multiple views is inherently more complex than single-view reasoning. The uncertainty-aware mechanism further complicates what should be a straightforward c→y process. Complexity typically means harder to interpret. That said, I'm not suggesting the proposed method doesn't make sense, but this seems to conflict with the paper's stated goal of "effectively solving the problem of opaque and untraceable reasoning processes in traditional models."

### Minor Concerns:
1. Line 141: "fundas" -> "fundus"
2. The generalization theory in Section 2.3 is commendable but could benefit from improved narrative flow. There’s a noticeable gap between the first paragraph of 2.3 and 2.3.1 that affects readability.

**Questions:**

Please check the weakness part.

---

> ### Author Response · Authors · 2025-11-25
> **Response to Reviewer eP2g [part 1]**
>
> **Response to w1:**
>
> Thanks for your suggestions. This conclusion is consistent with prior studies: although Transformers excel at modeling long-range global dependencies, they are relatively weak at capturing fine-grained features within small image patches. For example, Xu et al. [1] demonstrate that ViTs lack an intrinsic locality inductive bias, which leads to the loss of local detail information. In addition, Shamshad et al. and Khan et al. [2] provide systematic overviews of Transformer-based methods for medical diagnosis, and point out that, due to the absence of continuous local perception, these models face challenges in handling boundary and texture features in medical images. Several studies [3,4] further suggest that local continuity is essential for effective visual feature perception.
>
> **Table R1. Ablation study of backbones on MFIDDR. (Unit: %)**
>
> | Backbone  | Grading Acc. | Grading F1 | Concept AUPR | Concept F1 | Params (M) | Infer. (ms)  |
> | --------- | ------------ | ---------- | ------------ | ---------- | ------ | -------- |
> | VGG-16    | 85.63        | 85.26      | 67.69        | 66.88      | 15.29  | 6.93     |
> | ResNet-50 | 85.72        | 84.08      | 67.71        | 66.12      | 25.26  | 10.16    |
> | ViT-B     | 85.82        | 85.44      | 55.01        | 52.81      | 86.61  | 12.09    |
> | Swin v1-S   | 86.09        | 86.03      | 66.65        | 60.09      | 49.56  | 18.65   |
> | Swin v2-S   | 86.33      | 85.76      | 61.47      | 57.52      | 37.93 | 26.95  |
> | VMamba    | 83.03        | 82.29      | 59.43        | 52.15      | 14.60  | 9.27     |
> | **Ours**  | **86.75**    | **86.35**  | **72.26**    | **68.92**  | **6.70**   | 8.77 |
>
> From the quantitative results in Table R1, the ablation experiments confirm that our Hilbert-WKV is far superior to Transformer-based models in terms of fine-grained lesion concept perception and grading, and is more computationally efficient.
>
> References:
>
> [1]Vitae: Vision transformer advanced by exploring intrinsic inductive bias, Neruips, 2021.
>
> [2]Transformers in medical imaging: A survey, MIA, 2023.
>
> [3]Mamba: Linear-time sequence modeling with selective state spaces, COLM, 2024.
>
> [4]Vision-rwkv: Efficient and scalable visual perception with rwkv-like architectures, ICLR, 2024.
>
> **Response to w2:**
>
> In spatial feature maps, the motivation for introducing multi-directional RWKV is closely related to the locality-preserving advantages of space-filling curves such as the Hilbert curve. Some studies have shown that the Hilbert curve more effectively maps multidimensional data to one dimension while minimizing the likelihood that spatially adjacent points are mapped far apart in the serialized sequence. For example, prior work by Moon et al. [5] and Haverkort & van Walderveen [6] rigorously analyzed the clustering and locality-preserving behavior of different curves, showing that Hilbert consistently achieves lower dilation and clustering number than Morton (Z-order) or raster scans. Inspired by these findings, we argue that a single raster-like RWKV pass over a 2D feature map imposes a strong directional bias, so that vertically or diagonally neighboring pixels can become distant in the 1D sequence, effectively increasing dilation and degrading local clustering. By instead performing RWKV along multiple directions (e.g., left-to-right, right-to-left, top-to-bottom, bottom-to-top) and aggregating the resulting states, we emulate the locality-preserving spirit of Hilbert-like mappings: for a fixed receptive “window,” most neighboring pixels remain close in at least one traversal. This multi-directional design yields more isotropic modeling of local fundus structures, which is beneficial for downstream network modeling and local feature aggregation.
>
> In addition, we conduct comprehensive quantitative experiments (in Tables R1 and R2) comparing Hilbert-RWKV with its lightweight substitutes and various scanning strategies. The results show that Hilbert-RWKV achieves the best performance on the multi-view fundus imaging task across all metrics, further validating the advantage of its strong global modeling with locality-preserving design.
>
> **Table R2. Comparison of scanning strategies on MFIDDR. (Unit: %)**
>
> | Strategy   | Grading Acc. | Grading F1 | Concept F1 | Concept AUPR |
> | ---------- | ------------ | ---------- | ---------- | ------------ |
> | Sweep      | 86.29        | 85.94      | 65.24      | 69.61        |
> | Horizontal | 86.24        | 86.05      | 64.93      | 70.97        |
> | Vertical   | 86.10        | 85.96      | 64.75      | 70.46        |
> | Zigzag     | 86.24        | 85.70      | 63.89      | 67.60        |
> | **Ours**   | **86.75**    | **86.35**  | **68.92**  | **72.26**    |
>
>
> References:
>
> [5] Analysis of the clustering properties of the Hilbert space-filling curve, TKDE.
>
> [6] Locality-preserving properties of space-filling curves, Computational Geometry.

---

> > ### Author Response · Authors · 2025-11-25
> > **Response to Reviewer eP2g [part 2]**
> >
> > **Response to w3:**
> >
> > We appreciate your comment about OpenAI-TE3. In our setting, the textual inputs are not raw patient reports or electronic medical records, but a curated fundus-diagnosis knowledge corpus (e.g., lesion concept definitions, grading criteria, and guideline-style descriptions). These texts are therefore closer to fundus-specific professional English than to highly personalized clinical notes. For such knowledge-oriented understanding, OpenAI-TE3 produces satisfactory concept embedding features.
> >
> > Existing medical language embedding models can be roughly grouped into two families: BERT-based medical encoders (e.g., MedCPT [7], Clinical ModernBERT) and LLM-based medical embedding encoders (e.g., BioGPT [8], Me-LLaMA [9]). The former generally struggle with encoding long diagnostic knowledge passages (a limitation of BERT-based models), which substantially undermines the reliability of the resulting concept-text embeddings. The latter are primarily optimized for medical QA, and hidden features extracted from these LLM encoders suffer from practical adaptation issues such as inflexible feature dimensionality and unstable generation.
> >
> > For completeness, we evaluate alternative text encoders and report the results below:
> > | Method     | Grading Acc ↑ | Grading Kappa ↑ | Concept AUPR ↑ | Concept F1 ↑ |
> > |-----------|---------------|-----------------|----------------|-------------|
> > | MedCPT    | 86.31         | 75.01           | 71.38          | 66.54       |
> > | BioGPT| 86.20         | 74.78           | 70.32          | 66.15       |
> > | Me-LLaMA  | 86.03         | 74.36           | 68.09          | 65.34       |
> > | **Ours**  | **86.75**     | **76.05**       | **72.26**      | **68.92**   |
> >
> > The experimental results verify the strong capability of OPAI-TE3 in encoding long-form lesion concept texts.
> >
> > References:
> >
> > [7] Medcpt: Contrastive pre-trained transformers with large-scale pubmed search logs for zero-shot biomedical information retrieval, Bioinformatics, 2023.
> >
> > [8] BioGPT: generative pre-trained transformer for biomedical text generation and mining, Briefings in bioinformatics, 2022.
> >
> > [9] Me-llama: Foundation large language models for medical applications, Research square, 2024.
> >
> >
> >
> >
> > **Response to w4:**
> >
> > Incorporating text-based concept embeddings not only improves accuracy but also strengthens interpretability in two complementary ways.
> > (1) **Semantic anchoring of visual concepts.** Without text, concept vectors are purely data-driven and may encode spurious correlations between lesions and grades. By aligning each visual concept embedding with the embedding of its guideline-level textual definition (e.g., “hard exudates: yellowish deposits with sharp margins”), we explicitly constrain the learned concept space to remain close to clinically meaningful semantics. It allows clinicians to interpret each concept via its definition rather than as an abstract feature vector. Additionally, we include a visualization of concepts with and without text embeddings, which further shows that text guidance leads to more compact and semantically coherent concept clusters.
> >
> > (2) **Text-retrieval–based explanations.** Image concepts and diagnostic knowledge texts are embedded in a shared space, which allows us to retrieve the most similar guideline sentences or lesion descriptions for each lesion concept. In practice, this allows us to attach human-readable explanations such as: *“the model predicts MA because the image region is most similar to the description ‘small round red dots representing microaneurysms’.”* This explicit linkage between visual evidence, concept scores, and textual descriptions substantially improves transparency compared to using visual features alone.
> >
> > In summary, fusing visual concepts with text embeddings enforces consistency between concept representations and standardized diagnostic knowledge, and it further enables natural-language, guideline-level explanations for each predicted concept, thereby enhancing both the semantic clarity and the clinical usefulness of our explanations.

---

> > > ### Author Response · Authors · 2025-11-25
> > > **Response to Reviewer eP2g [part 3]**
> > >
> > > **Response to w5 and w9:**
> > >
> > > As for interpretability in the context of uncertainty modeling and multi-view reasoning, we reply as follows:
> > >
> > > (1) We do not deny that multi-view reasoning is more complex than single-view, as you pointed out. However, in many real-world applications, such as medical diagnosis (fundus imaging, CT), autonomous driving, and anomaly detection, multi-view setups are not only widely accepted in these domains but also crucial for significantly improving accuracy (DR grading: acc +8.32%). In the context of DR diagnosis, the most advanced fundus screening protocols already rely on multi-field fundus photography, where images are captured from four standard views to avoid blind spots that inevitably occur under a single viewing angle. We believe that an interpretable model is clinically meaningful only when it achieves both interpretability and accuracy. In addition, several medical studies [10,11,12] have already examined interpretability in multi-view and text–image multimodal settings, further underscoring the necessity of research in this direction.
> > >
> > > (2) **Interpretability of our dual-uncertainty-aware multi-view fusion.**
> > > $\hat{\mathbf{y}} = \sum_{i=1}^{V} \left[ W_c (1 - \Psi_v^{\mathrm{con}}) + (1 - W_c)(1 - \Psi_v^{\mathrm{gr}}) \right] \odot \hat{\mathbf{y}}^v$.
> > > While the verification of theoretical feasibility is complex, the fusion mechanism itself is intuitive. The interpretability of our approach comes from keeping the original, decomposable (c → y) reasoning chain intact, while augmenting it with an explicit, clinically meaningful “confidence scale” at each stage. Rather than fusing raw feature maps in a black-box manner, each view outputs its own lesion concepts and associated concept uncertainty. The multi-view fusion applies an explicit rule that assigns a higher weight to views with more confident, consistent concept predictions. Uncertainty thus serves as a quality indicator for both concepts and grades, allowing clinicians to see not only which views and concepts drive the final decision, but also how certain the model is about these underlying reasons. In summary, our method keeps the reasoning chain “view → concepts (with uncertainty) → grade (with uncertainty)” fully transparent, while using concept and grading uncertainty to perform a principled and interpretable multi-view fusion rather than a black-box ensemble.
> > >
> > > References:
> > >
> > > [10] Self-explainable AI for medical image analysis: A survey and new outlooks, Arxiv, 2024.
> > >
> > > [11] MERIT: Multi-view evidential learning for reliable and interpretable liver fibrosis staging, Medical image analysis, 2025.
> > >
> > > [12] Multimodal alignment and fusion: A survey, Arxiv, 2024.
> > >
> > > **Response to w6:**
> > >
> > > Lesion masks are generated by the HACDR-Net pretrained segmentation model on DDR (not MFIDDR and DRTID). After obtaining the segmentation masks for the four fundus views, we concatenate each mask with its corresponding image along the channel dimension to perform early fusion. The fusion feature $Img_f$ is calculated as follows,
> > > $Img_f = Concat(Pre(I_i), LC_i)$, here, the function $Pre(\cdot)$ is the preprocessing [13] in the DR image to reduce the influence of lighting conditions, $LC_i$ single-channel grayscale image of the input DR image.
> > >
> > > References:
> > >
> > > [13] APTOS: Eye Preprocessing in Diabetic Retinopathy, 2019.
> > >
> > > **Response to w7:**
> > >
> > > We have obtained approval from the hospital. If our paper is accepted, we will release the entire dataset together with all labels.
> > >
> > > **Response to w8:**
> > >
> > > We sincerely thank you for your suggestions. The incomplete discussion of the ablation studies in the main text (with part of the material only appearing in the appendix) is currently revising the manuscript to incorporate the experimental analysis you requested. Here, we attribute the slight decrease in concept prediction after introducing DU-MVFD to two main factors: (1) since our framework is multi-task, the checkpoint is selected according to the best grading performance, which occasionally leads to a model with marginally degraded concept accuracy. (2) DU-MVFD is most effective when the per-view concept and grading predictions are already reasonably accurate, so under the baseline MVCBM with low concept accuracy, adding DU-MVFD alone brings only modest gains. In contrast, the improved concept performance obtained when jointly using the Hilbert-RWKV and VT-RWKV modules highlights their strong ability to enhance the perception of fine-grained lesion concepts. We are currently working intensively to address the concerns you raised regarding the coherence of the methodology section, the depth of the interpretability analysis, and the discussion of the ablation studies. We invite you to download the revised manuscript shortly to review these updates.
> > >
> > > **Response to Minor Concerns:**
> > >
> > > Thank you for your reminders. According to your suggestions, we have made the necessary revisions to the manuscript.

---

> ### Comment · Reviewer_eP2g · 2025-11-28
>
> Thank you for your effort and detailed response. The authors perfectly addressed all of my concerns. I would like to raise my score and recommend accepting the paper.

---

> > ### Author Response · Authors · 2025-11-28
> > **Thank you again for your feedback and contrributions!**
> >
> > Thank you for your feedback and support. All clarifications and revisions will be reflected in the revision version. Thank you again for your help in improving the quality of our manuscript.

---

### Official Review · Reviewer_Dsqb · 2025-10-28

**Soundness:** 2
**Presentation:** 3
**Contribution:** 2
**Rating:** 2
**Confidence:** 5

**Summary:**

This paper introduces ProConMV, a provenance-enabled multimodal and multi-view framework for diabetic retinopathy diagnosis. The approach integrates lesion masks, GPT-generated clinical texts, and fundus images via a Hilbert-RWKV backbone, a Visual-Text RWKV reasoning module, and a dual uncertainty-aware fusion mechanism. While the framework is technically elaborate and theoretically motivated, the motivation and experimental design raise serious concerns. The multimodal data are synthetically constructed (segmentation-derived lesion masks, GPT-generated text, and simulated physician feedback), which undermines fairness, reproducibility, and clinical validity. Moreover, the model’s complexity appears disproportionate to the relatively simple five-class DR grading task, and the performance gains are marginal. The practical applicability of the proposed multi-view design to DR imaging is also questionable.

**Strengths:**

1. The framework integrates several advanced components, including RWKV attention mechanisms, uncertainty-aware fusion, and concept-level reasoning.
2. The authors provide a formal generalization bound for multi-view concept-based fusion, which adds some theoretical rigor.
3. The architecture is clearly presented, and the empirical results on public datasets are reasonable.

**Weaknesses:**

1. The multimodal inputs (lesion masks and clinical texts) are synthetically generated, raising major concerns regarding data validity, fairness, and reproducibility.
2. The paper provides no evaluation of segmentation accuracy or potential biases in the GPT-generated texts. Supervision leakage between modalities may possibly occur.
3. The proposed model is excessively complex for a relatively simple five-class DR grading task, yielding only marginal improvements in accuracy.
4. The ‘multi-view’ assumption is not well-grounded for fundus imaging, where multi-view acquisitions are uncommon in clinical settings. The method would be more convincing if validated on genuinely multi-view modalities (e.g., ultrasound, CT), which better reflect the intended design motivation.
5. The interpretability claims are largely theoretical, no user studies or physician-based validation are included to substantiate them.

**Questions:**

1. How was the accuracy of the generated lesion masks validated, and how sensitive are the results to segmentation errors?
2. Are the GPT-generated clinical texts consistent across samples with identical DR grades but differing lesion distributions?
3. Could the proposed approach be extended to genuinely multi-view medical imaging tasks, such as ultrasound or CT, where the ‘multi-view’ assumption naturally holds?

---

> ### Author Response · Authors · 2025-11-24
> **Response to Reviewer Dsqb [Part 1]**
>
> Thank you for your suggestions. We respond to your concerns as follows.
>
> **Response to W1:**
>
> Thank you very much for your profound insights. In the actual clinical process, the vast majority of fundus screening data do not have expert pixel-by-pixel lesion annotation or structured text reports. Therefore, the use of model-automatically extracted lesion masks and text enables its transferability and maintainability.
>
> 1) As for data validity, studies [1, 2] on fundus and related ophthalmic images have shown that when there is no pixel-level gold standard, using image-level labels, significant region localization, or weak annotations to generate automatic masks can stably improve diagnostic and localization performance and enhance interpretability. In addition, we have supplemented the table below with the methods that use lesion masks and added ablation experiments with and without lesion masks.
>
> | Method          | Lesion   |
> |-----------------|---------|
> | CrossFit        | ❌       |
> | ETMC            |❌|
> | MVCINN          | ❌ |
> | CVSRA-ViT       |❌ |
> | SMVDR           | ✅|
> | WMIMVDR         | ✅|
> |-----------------|---------|
> | Multi-Task        | ✅ |
> | MVCBM           | ✅ |
> | CEM (MV)        | ✅ |
> | PCBM (MV)       | ✅ |
> | SSMVCBM         | ❌  |
> | CLAT (MV)       | ❌  |
> | ProConMV (Ours) | –       |
>
> | Method     | Grading Acc ↑ | Grading Kappa ↑ | Concept AUPR ↑ | Concept F1 ↑ |
> |-----------|---------------|-----------------|----------------|-------------|
> | no lesion    | 85.21         | 73.98           | 68.59          | 66.26       |
> | with lesion| **86.75**     | **76.05**       | **72.26**      | **68.92**   |
>
> 2) As for fairness, the automatic generation process replaces the differences in subjective annotations by multiple doctors with a unified algorithm, reducing annotation deviations across institutions and populations.
>
> 3) As for reproducibility, compared to difficult-to-replicate human work, the generation pipeline of automatic masks and text is composed of publicly describable models and parameters, which can be independently replicated on public datasets. The segmentation model trained on the DDR dataset is HACDR-Net (in AAAI24). The code and weights are publicly available. The concept text includes information such as the definition, description, and diagnostic rules of the lesion. We will make public the prompt and specific text information generated by GPT for this (which will be in the appendix of the new paper).
>
>  Reference:
>
> [1]A deep learning system for predicting time to progression of diabetic retinopathy. Nature Medicine, 2024.
>
> [2]A Lesion-Fusion Neural Network for Multi-View Diabetic Retinopathy Grading. JBHI, 2025.
>
>
> **Response to W2:**
>
> Thank you for your questions. 1) As for "The Accuracy of Segmentation Masks" and "Supervised leakage", our process ensures that there is no supervision leakage: all the lesion segmentation masks used in this study were generated by a model that was pre-trained on an independent public dataset (DDR). This model was "frozen" after generating the masks, and its weights were never fine-tuned or re-trained on our main DR grading task dataset in any form. This practice of generating intermediate representations or masks using models pre-trained on independent datasets is very common in multimodal learning of medical images, such as [3]. 2) As for "The Potential Bias of GPT-Generated Text”, we agree with your concerns and hereby solemnly state that the fundus image dataset used in our experiments is not included in the training data of the GPT model. We use the GPT model to generate structured clinical text. Use a unified and standardized textual description as the semantic anchor shared by all samples, rather than generating customized descriptions for each sample. Therefore, the training data in our test set will not be leaked into the GPT model, and it will not be affected by text biases.
> In the revised edition of the paper, we elaborate on the above-mentioned process measures more clearly and incorporate discussions on the potential noise and limitations of the generative modality.
>
> Reference:
>
> [3] Multimodal Learning with Missing Modality: A Benchmark. CVPR, 2023.

---

> ### Author Response · Authors · 2025-11-24
> **Response to Reviewer Dsqb [Part 2]**
>
> **Response to W3:**
>
> We understand your concerns about the complexity and performance of the model, and would like to clarify the following: the increased complexity we have introduced is not redundant, but rather is aimed at achieving reliable, understandable and verifiable diagnostic decisions by doctors, which is crucial for clinical implementation. This work does not merely aim to achieve high classification accuracy. In the real clinical environment, diagnosis faces multiple challenges: i) The variation in image quality due to patient cooperation is a common occurrence. ii) Conflicts among information from different views and modals. A simple model cannot robustly handle these complexities. Therefore, the complexity of our model is designed to directly address these clinical real-world challenges.
>
> Our framework aims to build a fully process-interpretable system. From the perspective of reasoning mode, the concept bottleneck model (input->concept->result) is a recognized promising paradigm structure in the field of visual interpretability. Many research results [4] have shown that this type of model has constructed simple and intuitive interpretability methods compared to other posterior explanation models.
>
> We have extended the concept bottleneck paradigm to the multi-view domain and proposed multiple innovative points such as text and fusion explanations. The entire reasoning process is from multimodal input -> concept prediction of each view -> hierarchical prediction of each view -> multi-view fusion, which is clear and intuitive. From the perspective of dual uncertainty multi-view fusion, the theoretical feasibility proof of our theory is complex, but the fusion method is intuitive and easy to understand:
>
> $\hat{\mathbf{y}} = \sum_{i=1}^{V} \left[ W_c (1 - \Psi_v^{\mathrm{con}}) + (1 - W_c)(1 - \Psi_v^{\mathrm{gr}}) \right] \odot \hat{\mathbf{y}}^v$.
>
> $w$ represents the model parameters, with the preceding $\Psi_v^{\mathrm{con}}$ representing the conceptual uncertainty and the latter $\Psi_v^{\mathrm{gr}}$ representing the graded uncertainty. This design for view uncertainty can also be extended to low-quality multi-view classification tasks. Each step's "complexity" is designed to transform the "input-output" black box into a pathological logic that doctors can understand and verify.
>
> More importantly, while achieving interpretable diagnostic results, the accuracy of our model's grading also shows a significant improvement. As shown in the following two tables.
>
> Table 1. DR grading on MFIDDR
>
> | Method              | MFIDDR Acc.↑ | MFIDDR Spe.↑ | MFIDDR Kap.↑ | MFIDDR F1↑ | Infer. Time ↓ |
> |---------------------|--------------|--------------|--------------|------------|----------------|
> | CrossFit            | –            | –            | –            | –          | –              |
> | ETMC                | 81.54        | 83.44        | 64.76        | 79.74      | **6.61**       |
> | MVCINN              | 80.10        | 83.32        | 62.45        | 78.86      | 31.31          |
> | CVSRA-ViT           | 82.61        | 86.77        | 68.57        | 81.94      | 71.53          |
> | SMVDR               | 84.01        | 91.30        | 71.36        | 83.69      | 65.71          |
> | WMIMVDR             | 84.15        | 89.95        | 71.16        | 83.59      | 25.44          |
> | Multi-Task          | 83.73        | 89.06        | 70.24        | 83.12      | 8.24           |
> | MVCBM               | 83.22        | 88.22        | 69.12        | 82.43      | 19.67          |
> | CEM (MV)            | 84.12        | 88.77        | 70.83        | 83.45      | 21.25          |
> | PCBM (MV)           | 83.52        | 91.19        | 70.35        | 83.29      | 17.56          |
> | SSMVCBM             | 82.75        | 85.81        | 67.55        | 81.51      | 20.71          |
> | CLAT (MV)           | 82.89        | 86.66        | 68.16        | 81.88      | 33.02          |
> | **ProConMV (Ours)** | **86.75**    | **92.79**    | **76.05**    | **86.35**  | 8.77           |
>
> Table 2. DR concept classfication on MFIDDR
>
> | Method              | MFIDDR AUPR↑ | MFIDDR ACC↑ | MFIDDR F1↑ | MFIDDR RL↓ | MFIDDR HL↓ |
> |---------------------|-------------:|------------:|-----------:|-----------:|-----------:|
> | Multi-Task          | 54.69        | 93.87       | 51.69      | 3.73       | 5.54       |
> | MVCBM               | 61.56        | 94.22       | 59.10      | 3.21       | 5.36       |
> | CEM (MV)            | 65.47        | 94.74       | 60.42      | 2.86       | 5.21       |
> | PCBM (MV)           | 68.12        | 94.85       | 66.08      | 1.91       | 4.96       |
> | SSMVCBM             | 66.25        | 94.42       | 63.34      | 2.17       | 5.02       |
> | CLAT (MV)           | 63.89        | 94.63       | 59.15      | 2.98       | 5.45       |
> | **ProConMV (Ours)** | **72.26**    | **95.42**   | **68.43**  | **1.45**   | **4.47**   |
>
> Reference:
>
> [4] The decoupling concept bottleneck model, TPAMI, 2024.

---

> ### Author Response · Authors · 2025-11-24
> **Response to Reviewer Dsqb [Part 3]**
>
> **Response to W4:**
>
> Thanks. Your question about the widespread application of multi-view fundus imaging in clinical practice has prompted us to elaborate further on the clinical basis and research purpose of this study. You mentioned that multi-angle acquisition is not common, and this observation is in line with the current situation in the basement large-scale screening scenario. However, this precisely points to the core flaw of the current DR screening system. A standard single-view fundus image typically captures only a $45^\circ$ or $50^\circ$ field of view, which corresponds to approximately 12.5\% of the retinal area. This limited coverage is insufficient for detecting lesions that occur outside the central view, such as peripheral microaneurysms and neovascularization. Clinical studies [5,6] have shown that single-view non-mydriatic fundus photography fails to meet technical requirements for DR grading in real-world practice, thereby constraining its clinical utility. To address this shortcoming, multi-view retinal imaging has been proposed as a more comprehensive alternative, offering wider spatial coverage and significantly improved lesion detection sensitivity. Our design intention is to develop an AI system that matches the gold standard requirements for clinical diagnosis.
>
> Reference:
>
> [5] Single-field non-mydriatic fundus photography for diabetic retinopathy screening: a systematic review and meta-analysis, Ophthalmic Research, 2019.
>
> [6] Efficacy of single-field non-mydriatic digital fundus photography for screening diabetic retinopathy. Journal of the Korean Ophthalmological Society, 2011.
>
> **Response to W5:**
>
> Thank you for your important question. We agree that large-scale physician user studies will be an important extension in the future. However, at this stage, we emphasize as full-process explainability is a systematic design, with the core being to align the model's reasoning with the diagnostic logic of clinical doctors at a deep level. The claim of explainability is based on three solid foundations. 1) Our concept reasoning framework is based on the concept reasoning units recognized by Wen et al. [7] and other works, making the decision-making logic transparent and traceable. 2) Our explainability is not an afterthought but is integrated from the architectural level with the prior knowledge of doctors. And a doctor intervention section has been specially designed, leaving an entry for human experts to correct and confirm during the reasoning process. 3) The model not only outputs the final grading but also generates lesion concept activation maps, the confidence of each concept, and the uncertainty of concepts and grading in multiple views.
>
> Thank you for giving us the opportunity to clarify. We will clarify this position in the revised version of the paper and plan to include physician evaluation as a core task in the future.
>
> Reference:
>
> [7] Concept-based lesion aware transformer for interpretable retinal disease diagnosis. IEEE Transactions on Medical Imaging, 2024.
>
> **Response to Q1:**
>
> Thanks for your comments. In regards to the comment ”How was the accuracy of the generated lesion masks validated?”, the lesion masks are extracted from the original fundus images using HACDR-Net [8]. We utilized the public datasets IDRID and DDR to pre-train a multi-lesion segmentation model as the lesion extractor. The accuracy of our automatically generated lesion masks was validated through a combination of established metrics and clinical correlation.
>
> Moreover, as for the segmentation errors, we clarify that error segmentation will have a small impact on the performance of classification. Firstly, the purpose of our Lesion Extractor is to obtain an approximate location range of possible lesions through lesion masks, and it does not require strict segmentation results. Furthermore, our model is designed to significantly reduce the bias introduced by incorrectly segmented masks. As a multi-view learning model, obtaining lesion masks from four views significantly enhances the credibility of lesion segmentation in fundus images compared to a single view. Additionally, we designed the Hilbert RWKV Block, and fusing it with the RGB image channels can mitigate the impact of erroneous lesion masks on the prediction results. We conducted experiments on the issue of sensitivity. Specifically, we incorporated low-quality (with a Dice score decrease of 7.6% in DDR) lesion masks into the inference process. The experimental table is as follows:
>
> | Method     | Grading Acc ↑ | Grading Kappa ↑ | Concept AUPR ↑ | Concept F1 ↑ |
> |-----------|---------------|-----------------|----------------|-------------|
> | weak mask    | 85.46         | 75.52          | 70.89          | 67.76     |
> | strong mask | **86.75**     | **76.05**       | **72.26**      | **68.92**   |
>
> Reference:
>
> [8]HACDR-Net: Heterogeneous-Aware Convolutional Network for Diabetic Retinopathy Multi-Lesion Segmentation, AAAI, 2024.

---

> > ### Author Response · Authors · 2025-11-24
> > **Response to Reviewer Dsqb [Part 4]**
> >
> > **Response to Q2:**
> >
> > Thank you for giving us the opportunity to clarify. In this work, we provided a unified textual description for all samples as a shared semantic anchor point, rather than generating customized texts for individual samples. The aim was to lay a stable and reliable semantic foundation for multimodal learning. Specifically, the design of using a unified text was based on two core considerations. First, in multimodal learning, the unified text serves as a constant semantic target, which can guide the visual model to focus on key visual features and avoid being interfered by the subtle differences in lesion distribution among samples, helping the model learn robust cross-modal associations. Second, it allows for complete control over the quality of text features, avoiding additional noise such as hallucinations, biases, or inaccuracies that may occur when GPT generates personalized descriptions, providing a clean and reliable semantic input for the system. This mechanism works closely with the overall framework, aiming to guide the model through standardized text semantics to focus on the key visual concepts of DR diagnosis, enhancing the robustness and interpretability of multimodal learning, rather than using text to repeat specific visual details.
> > We will clearly describe this mechanism in the methodology section of the paper to avoid potential misunderstandings. The conceptual text contains information such as the definition, description and diagnostic rules of the lesion. We will make public the prompt used by GPT to generate this information as well as the specific text content (it will be included in the appendix of the new paper).
> >
> >
> > **Response to Q3:**
> >
> > Thank you for this insightful question. We strongly believe that the core principles of our framework are highly generalizable and can be directly extended to other genuinely multi-view medical imaging tasks, such as ultrasound or CT. The strength of our approach lies in its abstraction from the specific imaging modality and its focus on a problem-agnostic fusion and reasoning philosophy. While our current implementation and evaluation are focused on multi-view fundus photography (i.e., a domain with a critical, unmet clinical need), the proposed framework is architected for generality. The pillars of uncertainty-aware multi-view fusion, concept-based reasoning, and human-in-the-loop collaboration are fundamental to tackling the challenges of robust and interpretable AI in virtually any multi-view or multi-modal medical imaging context. We view this generalizability as a key strength of our work and plan to pursue these exciting extensions in future research.

---

> ### Author Response · Authors · 2025-11-28
> **Follow-up on our rebuttal**
>
> Dear Reviewer Dsqb,
>
> We have provided our response to your initial review and revised the manuscript to address each of your concerns, and the code will also be open source to increase transparency. With the discussion period ongoing, we would be grateful if you could spare a moment to review our rebuttal. We stand ready to answer any further questions or offer additional clarifications as required.
>
> Your insights will be invaluable in refining the final version of our work. Thank you once again for your careful time and valuable efforts!
>
> Thanks,
>
> Authors of submission 13076

---

### Official Review · Reviewer_pmak · 2025-10-31

**Soundness:** 3
**Presentation:** 3
**Contribution:** 3
**Rating:** 8
**Confidence:** 4

**Summary:**

ProConMV is a provenance-enabled, concept-bottleneck framework for multi-view DR diagnosis. Contributions: (i) *Hilbert-RWKV*, a linear-time vision backbone that preserves 2D locality via Hilbert-curve tokenization; (ii) *VT-RWKV* to align visual lesion concepts with textual clinical descriptions (fixed, LLM-derived) and enable clinician test-time intervention; (iii) a *multi-view CBM generalization bound* motivating dynamic fusion: with per-view predictions $\hat{y}^{(v)}$ and weights $w_v$ ($\sum_v w_v=1$), the late-fusion $\hat{y}=\sum_v w_v \hat{y}^{(v)}$ admits a tighter upper bound when
$$
\mathrm{Cov}\big(w_v,\ \ell_y(G(c^{(v)}),y)\big)\le 0
\quad\text{and}\quad
\mathrm{Cov}\big(w_v,\ | \hat{c}^{(v)}-c^{(v)}|_1\big)\le 0.
$$
Experiments on MFIDDR (4 views) and DRTiD (2 views) report SOTA grading and concept prediction with fast inference.

**Strengths:**

* **S1: Principled fusion for multi-view CBMs.** The bound explicitly exposes covariance terms, giving a clean criterion for dynamic, reliability-aware fusion instead of ad hoc averaging.

* **S2: Efficient locality-aware backbone.** Hilbert-RWKV is a clear, linear-time adaptation of RWKV to images with locality preserved by the Hilbert curve; ablations show gains over ResNet/ViT/VMamba.

* **S3: End-to-end interpretability.** Concept-level predictions align with clinical findings and support test-time clinician intervention; uncertainty provides verifiable provenance.

* **S4: Solid empirical results.** Consistent improvements on grading and lesion concepts with competitive latency; ablations attribute gains to each module.

**Weaknesses:**

* **W1: Inputs at inference are under-specified.** The abstract/figures imply *Seg* masks and "clinical text." Section 2.1.4 suggests the text is *fixed, offline* lesion descriptions (not patient notes). Clarify precisely what modalities are *required at test time*: images only? images+seg masks? any patient text?

* **W2: Theory $\to$ practice bridge is assumed.** The bound requires negative correlation between $w_v$ and *loss/error*, but the implementation weights by *uncertainty*. Please verify (with plots/statistics) that per-view uncertainties are positively correlated with per-view grading loss and concept error, so Eq. (16) instantiates the bound's prescription.

* **W3: Context vs. locality-preserving Transformers.** No Swin/hierarchical-window baseline is reported. A brief comparison (accuracy/latency or discussion) would better situate Hilbert-RWKV among locality-aware designs.

* **W4: Reproducibility of text embeddings.** Provide the LLM prompts, curation steps, and sensitivity to paraphrases; VT-RWKV depends on these fixed descriptions.

* **W5: Minor notation/editorial.** Eq. (17) mixes $\alpha/\lambda$ and inconsistent indices; clarify focal-loss form (multi-class vs. one-vs-rest). Correct "GPT-5 (Achiam et al., 2023)" to GPT-4.

**Questions:**

1. **Exact test-time inputs:** Are lesion *segmentation masks* provided at inference? Is any *patient-specific* text used, or only fixed lesion descriptions?

2. **Uncertainty vs. error:** Show quantitative correlations (scatter + Spearman/Pearson) between per-view uncertainty and (a) per-view grading loss, (b) per-view concept error; ideally on validation data.

3. **Backbone context:** How does Hilbert-RWKV compare to Swin (accuracy/latency) on your tasks? Any trade-offs you can articulate?

4. **Text prompts:** Release prompts and an ablation on rephrasing quality to establish robustness of VT-RWKV.

5. **Loss details:** Specify the exact focal-loss variant and the tuned weighting between grading and concept losses.

---

> ### Author Response · Authors · 2025-11-27
> **Response to Reviewer pmak [part 1]**
>
> Thank you for your positive evaluation and valuable suggestions. Regarding your concerns about the effectiveness of our backbone within locality-aware designs compared to hierarchical-window baselines, we have addressed this by adding comparative experiments with Swin Transformer. We have also verified the positive correlation between per-view uncertainties and per-view grading loss and concept error with plots. In addition, we clarified all previously under-specified information in the manuscript and carefully checked all equations and citations for correctness. Below we provide clarifications to the expressed concerns.
>
> 1. **Exact test-time inputs (W1, Q1).**
>
>    Thank you for giving us the opportunity to clarify. At test time, our model only requires the fundus images as input. The lesion masks used in our framework are generated by HACDR-Net [1], which is pretrained on the DDR dataset [2], so no additional manual annotations are needed.
>
>    Regarding the text modality, we adopts fixed, offline clinical guideline–style lesion descriptions rather than personalized patient notes. This design choice is motivated by two considerations. First, using a unified textual description provides a stable and consistent semantic anchor for multimodal learning, allowing the visual model to focus on core lesion characteristics without being influenced by case-specific lesion variations. This strengthens the robustness of cross-modal alignment. Second, unified text ensures full control over textual quality and avoids potential noise such as hallucinations, biases, or inaccuracies that may arise from generating patient-specific descriptions. In practice, real clinical deployment scenarios typically provide only fundus images, without structured diagnostic reports, making unified clinical descriptions a more realistic and reliable choice.
>
>    We sincerely appreciate your comment. In the revised manuscript, we will clarify the required inputs during inference as well as the role of the clinical text used in our framework.
>
>    Reference:
>
>    [1] HACDR-Net: Heterogeneous-Aware Convolutional Network for Diabetic Retinopathy Multi-Lesion Segmentation, AAAI, 2024.
>
>    [2] Diagnostic assessment of deep learning algorithms for diabetic retinopathy screening, Information Sciences, 2019.
>
> 2. **Correlation between uncertainty and error (W2, Q2).**
>
> Thank you for your valuable comments. As for the question about uncertainty vs. error, according to your suggestions, we provide scatter-plot visualizations of concept confidence (concept confidence = 1 - concept uncertainty) versus concept loss, and of grading confidence versus grading loss in **Fig. 3 and Fig. 7(a) of the revised manuscript**. Through scatter plots and Pearson correlation coefficient $\rho$, the correlation between the graded loss of a single view and the corresponding dual uncertainty fusion weights is visually demonstrated. If $\rho$ is negatively correlated, it indicates that the view with a higher graded loss (i.e., the less reliable view) is assigned a lower fusion weight, while the opposite is true, with a higher weight being assigned. This directly verifies the rationality of the weight allocation logic. In summary, considering the above observations and the confidence–loss distributions across views, we can conclude that grading confidence and loss are negatively correlated.

---

> ### Author Response · Authors · 2025-11-27
> **Response to Reviewer pmak [part 2]**
>
> 3. **Comparison with Swin/hierarchical-window Transformers (W3, Q3).**
>
>    Thank you for the suggestion. We have added comparisons between Swin Transformer (Swin v1-S and Swin v2-S) and our method, with the results presented in Table 1 below:
>
>    **Table 1. Ablation study of backbones on MFIDDR. (Unit: %)**
>
>    | Backbone  | Grading Acc. | Grading F1 | Concept AUPR | Concept F1 | Params (M) | Infer. (ms) |
>    | --------- | ------------ | ---------- | ------------ | ---------- | ---------- | ----------- |
>    | VGG-16    | 85.63        | 85.26      | 67.69        | 66.88      | 15.29      | 6.93        |
>    | ResNet-50 | 85.72        | 84.08      | 67.71        | 66.12      | 25.26      | 10.16       |
>    | ViT-B     | 85.82        | 85.44      | 55.01        | 52.81      | 86.61      | 12.09       |
>    | Swin v1-S | 86.09        | 86.03      | 66.65        | 60.09      | 49.56      | 18.65       |
>    | Swin v2-S | 86.33        | 85.76      | 61.47        | 57.52      | 37.93      | 26.95       |
>    | VMamba    | 83.03        | 82.29      | 59.43        | 52.15      | 14.60      | 9.27        |
>    | **Ours**  | **86.75**    | **86.35**  | **72.26**    | **68.92**  | **6.70**   | 8.77        |
>
>    As shown above, although Swin Transformers achieve grading performance close to ours, they still fall slightly behind overall. Moreover, for lesion concept prediction, Swin models perform noticeably worse in both AUPR and F1 compared with our method. In addition, our method uses fewer parameters and achieves faster inference, demonstrating clear advantages in efficiency and locality-aware structural modeling. Specifically, our model reduces parameter count by approximately 6–7× and improves inference speed by about 2–3× compared with Swin Transformers, further highlighting its efficiency and stronger capability in capturing local structural information. These results collectively demonstrate the strength of the proposed Hilbert-RWKV design among locality-aware architectures.
>
> 4. **Design and Robustness of Text Prompts for VT-RWKV (W4, Q4).**
>
>    Thank you for bringing this up. All DR lesion concept texts used in our framework are obtained through a six-round dialogue with GPT-4, where each round extracts one core attribute of the lesion, including its typical fundus appearance, key visual characteristics, and its correspondence to different DR stages. The final concept descriptions combine guideline-style clinical knowledge while avoiding any patient-specific information, thus completely eliminating the risk of cross-modal leakage. We provide the full prompt design and GPT outputs in **Fig. 10 of the revised manuscript**.
>
>    **Table 2. Ablation on Textual Concepts.**
>
>    | Text Setting  | Grading Acc. | Concept F1 |
>    | ------------- | ------------ | ---------- |
>    | no text       | 0.8568       | 0.6439     |
>    | weak text     | 0.8603       | 0.6690     |
>    | complete text | **0.8675** | **0.6843** |
>
>    To evaluate the sensitivity of VT-RWKV to text quality and paraphrasing, we conducted ablation experiments using no text and weak text variants. As shown above, compared with the complete-text setting, the weak-text variant yields only small drops of 0.72% in grading accuracy and 1.53% in concept F1, indicating strong robustness of VT-RWKV to paraphrasing or reduced text richness. In the no-text setting, grading accuracy and concept prediction drop by 1% and 4%, respectively, demonstrating that text is beneficial but the model is not overly sensitive to any specific prompt phrasing.
>
>    These results confirm that the design of fixed lesion descriptions is stable, and robust to paraphrase variations, supporting the reliability of VT-RWKV in multimodal concept learning.

---

> > ### Author Response · Authors · 2025-11-27
> > **Response to Reviewer pmak [part 3]**
> >
> > 5. **Editorial, Focal Loss Details and Loss Balancing (W5, Q5).**
> >
> >    Thank you for mentioning this. We have corrected Eq. (17) in the revised manuscript and systematically checked all equations throughout the paper for consistency and accuracy. Regarding the loss formulation, we adopt a one-vs-rest focal loss for DR grading. Although DR grading is a multi-class task, the final prediction layer treats each class as an independent binary decision, which aligns with common practice for handling class imbalance.
> >
> >    As described in Sec. 2.4 of the main manuscript, the first term in the loss corresponds to the focal loss (with focusing parameter $γ$) for class-imbalanced DR grading, while the second term is a binary cross-entropy loss for concept prediction. The balancing coefficient $α$ adjusts the relative contribution of the two losses. By jointly optimizing both terms, the model is encouraged to learn faithful concept representations while improving the final grading performance.
> >
> >    To clarify the weighting between the grading and concept losses, we provide a two-dimensional hyperparameter ablation in **Appendix A.6**. In this analysis, $α$ controls the weight of the concept prediction loss, while $γ$ is the focusing parameter of the focal loss for handling class imbalance in DR grading. As shown in **Fig. 8 of the revised manuscript**, increasing α enhances concept prediction performance but leads to a decrease in grading accuracy, which is inconsistent with our objective of prioritizing DR grading. By balancing this trade-off, we determine $α=0.2$ and $γ=2.0$ as the optimal settings.

---

> ### Author Response · Authors · 2025-11-28
> **Follow-up on our rebuttal**
>
> Dear Reviewer pmak,
>
> We have provided our response to your initial review and revised the manuscript to address each of your concerns. With the discussion period ongoing, we would be grateful if you could spare a moment to review our rebuttal. We stand ready to answer any further questions or offer additional clarifications as required.
>
> Your insights will be invaluable in refining the final version of our work. Thank you once again for your careful time and valuable efforts!
>
> Thanks,
>
> Authors of submission 13076

---

### Official Review · Reviewer_TMiz · 2025-10-31

**Soundness:** 2
**Presentation:** 3
**Contribution:** 2
**Rating:** 6
**Confidence:** 3

**Summary:**

The paper builds a multi-view DR grading system that tries to follow a clinician’s workflow: first detect lesion concepts on each fundus view, then map those concepts through a guideline-style reasoning chain to a grade, and finally fuse views by letting more reliable evidence speak louder. Concretely, it uses a lightweight Hilbert-RWKV backbone for local concept perception, a visual–text interaction module to align images with lesion terms, and a dual-uncertainty scheme (at the concept and grade levels) to drive dynamic fusion; the pipeline is provenance-aware so a reviewer or clinician can correct a concept at test time and see the final grade update accordingly. The authors also argue—via a simple generalization bound—that assigning higher weights to lower-loss/uncertainty views is preferable to static averaging. On MFIDDR and DRTiD, the method reports better grading and concept recognition than standard multi-view baselines, with ablations supporting the contribution of the backbone, the vision–text alignment, and the uncertainty-aware fusion, and with favorable efficiency numbers.

**Strengths:**

A clear, provenance-aware pipeline that mirrors clinical practice: detect lesion concepts per view, reason via guideline-like rules, then fuse views. The evidence chain is exposed end to end, so test-time corrections to concepts transparently update the final grade—useful for auditing and real clinical use.
The dynamic fusion is principled and easy to grasp: concept- and grade-level uncertainties steer weights toward more reliable views, with a simple bound to justify why this should beat static averaging. Ablations largely support the mechanism, not just the outcome.
Thoughtful engineering with broad empirical coverage. A lightweight backbone and vision–text interaction keep inference efficient, and results on two multi-view datasets—plus intervention and ablation studies—make a convincing case for utility when views are noisy or partially informative.

**Weaknesses:**

Dynamic fusion relies on uncertainty to allocate weights, but the article does not fully demonstrate that these uncertainties are well calibrated and consistent across perspectives. If the confidence level is too high or too low, or if the confidence scales from different perspectives are inconsistent, the fusion process may actually amplify the noise, ultimately affecting the credibility and reproducibility of the conclusion. It is suggested to supplement temperature calibration here ECE/ACE or risk coverage curve and cross perspective consistency analysis.

The author's theoretical motivation is that "weight should be negatively correlated with loss/uncertainty", but this lacks direct empirical evidence and counterfactual comparison. A more ideal approach would be to provide a statistical analysis of the correlation coefficient between weights and losses (or uncertainties) during the training process over iterations, and to design a comparison of scrambled uncertainties or using uniform weights to demonstrate how performance degrades when the current lift is disrupted, thus truly closing the chain of "why it works".

In terms of practicality, stronger clinical transferability and interactive adaptation evidence may be needed, such as robustness in out of domain settings (different devices, hospitals, imaging conditions); Degradation curve and automatic weight reduction capability when there is a lack of perspective or poor perspective quality; And interactive intervention experiments involving real readers (time constraints, net benefits/costs of one intervention, reader consistency). These will determine whether the method can be smoothly embedded into the actual workflow, rather than just performing well within the dataset.

**Questions:**

The uncertainty story is not fully convincing. I would like pre-/post–temperature ECE, ACE, and risk–coverage, and a cross-view consistency check: map each view’s confidence to the same calibration curve and compare, to confirm they are on the same scale and won’t bias fusion weights due to scale differences.

The “higher uncertainty → lower weight” mechanism also needs more direct evidence. A formal theorem is not necessary, but please provide a simple diagnostic: correlations between fusion weights and losses/uncertainties during training (or at convergence). Add two controls: shuffle the uncertainties, and enforce uniform weights. If performance drops under these controls, the mechanism assumption is more credible.

Please clarify transferability and robustness. Provide a small out-of-domain result and degradation curves for missing or low-quality views (randomly dropping 1–N views or adding real noise/occlusion), and show whether the model automatically down-weights such views. Also explain the source and order of the clinical text and concept labels, and run a weak/no-text ablation (remove or weaken text; report text-only and image-only ceilings) to rule out template or closed-loop labeling shortcut effects.

---

> ### Author Response · Authors · 2025-11-26
> **Response to Reviewer TMiz [part 1]**
>
> **Response to W1 & Q1:**
>
> We sincerely appreciate your recognition of the motivation, novelty, and research value of our work. Regarding your questions on the effectiveness and robustness of our core module, we have provided extensive experimental results to address them. Notably, except for the weak/untext ablation studies, all experimental results are computed using the final optimal model weight (solely for inference). It ensures a fair comparison, as retraining under noisy conditions could compromise the fairness of the results. We provide our responses from the following aspects:
>
> **Table A1. View-level grading accuracy in our model.**
>
> | View  | Grading Accuracy in four views |
> |:-----:|:-----------------------------:|
> | View 1 | 0.8061 |
> | View 2 | 0.8103 |
> | View 3 | 0.8001 |
> | View 4 | 0.7834 |
> | View 1+2| 0.8232 |
> | View 1+2+3| 0.8491 |
> | View 1+2+3+4| 0.8675 |
>
> **Table A2. Sample-wise multi-view consistency statistics.**
>
> | Metric                                   |Grading  Value                |
> |------------------------------------------|---------------------:|
> | Average fraction of correct views / sample | 0.8000            |
> | Samples with **all** views correct          | 0.6160            |
> | Samples with **≥ 3** views correct          | 0.7536|
> | Samples with **≥ 2** views correct          | 0.8759|
> | Samples with **≥ 1** view correct           | 0.9544            |
>
>
> **Table A3. Pairwise agreement between views (grading prediction agreement rate).**
>
> | View pair        | Agreement |
> |:----------------:|----------:|
> | View 1 vs View 2| 0.8047    |
> | View 1 vs View 3 | 0.7861    |
> | View 1 vs View 4 | 0.7759    |
> | View 2 vs View 3 | 0.7829    |
> | View 2 vs View 4 | 0.7768    |
> | View 3 vs View 4 | 0.7578    |
>
> In practice, the four views in retinal imaging captured from different angles typically exhibit high consistency in terms of quality, resolution, and content. To validate this in our dataset, we plot scatter diagrams of the confidence scores of the four views produced by our model (directly computed from the best-performing model) versus the grading loss in **Fig. 3 and Fig. 7 of the revised manuscript**. Furthermore, we provide the grading accuracy and the complete consistency analysis data for each view. From these results, we make three observations:
>
> (1) The confidence distributions of all views are similar, concentrated in the range 0.15–0.4 (0.25 is the average).
>
> (2) View 2 attains slightly higher grading accuracy than the other views, but the accuracy gap across views is below 2%, indicating that there is no multi-view imbalance issue.
>
> (3) Overall, the grading loss is negatively correlated with the confidence. Comparing different views, the view with lower loss (view 2) also exhibits higher confidence, which is consistent with negative correlation. In addition, compared with conventional average late fusion, our method improves the grading accuracy by about 1%.
>
> These experimental results confirm that the multi-view design is reasonable and that inherent differences in scale and other factors do not induce view imbalance in confidence, loss, or other view-wise statistics.

---

> > ### Author Response · Authors · 2025-11-26
> > **Response to Reviewer TMiz [part 2]**
> >
> > **Response to W2 & Q2:**
> >
> > 1) Specifically, to remedy the lack of evidence for the “higher uncertainty → lower weight” mechanism raised in your comments, we provide scatter-plot visualizations of concept confidence (concept confidence = 1 - concept uncertainty) versus concept loss, and of grading confidence versus grading loss in **Fig. 3 and Fig. 7(a) of the revised manuscript**. Through scatter plots and the Pearson correlation coefficient $\rho$, the correlation between the graded loss of a single view and the corresponding dual uncertainty fusion weights is visually demonstrated. If $\rho$ is negatively correlated, it indicates that the view with a higher graded loss (i.e., the less reliable view) is assigned a lower fusion weight, while the opposite is true, with a higher weight being assigned. This directly verifies the rationality of the weight allocation logic. In summary, considering the above observations and the confidence–loss distributions across views, we can conclude that grading confidence and loss are negatively correlated.
> >
> > 2) As for the suggestions that adding two controls: shuffle the uncertainties, and enforce uniform weights. We have added the following experimental results.
> >
> > **Table A4: DR Grading and Concept Prediction Performance Comparison with Control Groups (Shuffled Uncertainties/Uniform Weights)**
> >
> > | Method             | Grading ACC. | Grading Kappa | Concept Pred. AUPR | Concept Pred. F1 |
> > |--------------------|-------------|---------------|---------------------|------------------|
> > | Baseline (Concat)  | 86.23       | 75.23         | 70.69               | 67.89            |
> > | Late Fusion (Avg.) | 86.17       | 75.24         | 71.30               | 67.87   |
> > | TMC                | 86.01       | 74.58         | 70.45               | 67.24            |
> > | Moe                | 86.52       | 75.29         | 71.37               | 68.26            |
> > | Ours               | 86.75       | 76.05         | 72.26               | 68.43            |
> >
> > By comparing the performance of Ours with that of the control group, when uncertainty is disrupted, the relationship between weights and loss/uncertainty is broken, and the performance (such as ACC/Kappa, etc.) declines. After forcing uniform weights, the weights do not adjust according to loss/uncertainty, and the performance is worse than Ours. These experimental results once again demonstrate that the strategy of integrating with dual uncertainties is more effective and reliable.

---

> ### Author Response · Authors · 2025-11-26
> **Response to Reviewer TMiz [part 3]**
>
> **Response to W3 & Q3:**
>
>
> 1) In response to your suggestion of adding Gaussian noise to simulate low-quality views, we add 30% and 50% noise into the best-performing view (view 2), and report the results in Table A6 and in **Fig. 7 of the revised manuscript**. Even though the accuracy of view 2 drops substantially, the overall model performance decreases by only 2–4% and even surpasses existing models without noise corruption. Moreover, in **Fig. 7 of the revised manuscript**, the confidence distribution of the noisy view 2 changes drastically compared with the noise-free case in **Fig. 7(a) of the revised manuscript**: most samples now have confidence in the range 0–0.15 instead of 0.15–0.4, further illustrating the strong negative correlation between confidence and loss. These findings provide evidence for the model's efficacy and robustness on low-quality fundus images, as well as the soundness of the dynamic fusion approach.
>
> **Table A5. Ablation on Textual Concepts.**
>
> |Text Setting| Grading acc. |Concept F1.|
> |:----------------:|----------:|----------:|
> |no text |0.8568|0.6439|
> | weak text |0.8603   |0.6690    |
> | complete text | **0.8675**    |**0.6843**    |
>
> **Table A6. Robustness to noisy views (Gaussian noise injected into view 2).**
>
> | Method | Noise in v2 | Grading ACC | Grading Kappa |
> |:------:|:-----------:|------------:|----------:|
> | MVCBM  | +0%         | 0.8322      | 0.6912    |
> | **Ours**   | +0%         | **0.8675**  | **0.7605**  |
> | MVCBM  | +30%        | 0.7869      | 0.6435    |
> | **Ours**   | +30%        | **0.8487**  | **0.7331**  |
> | MVCBM  | +50%        | 0.7456      | 0.6079    |
> | **Ours**   | +50%        | **0.8213**  | **0.7027**  |
>
> 2) In the no-text setting, i.e., the ablation in **Table 3 of the revised manuscript** where VT-RWKV is disabled (marked with a cross), the grading and concept accuracies drop by 1% and 2%, respectively. We clarify that the textual concepts are obtained from GPT as two-sentence descriptions for each lesion concept, and do not contain any patient-specific information (thus avoiding modality leakage). In Table A5 and A6 We provide ablation experiments with weaker textual descriptions, where the grading accuracy and concept F1 decrease by 0.72% and 1.53% compared with our full-text setting. The exact textual prompts and descriptions are provided in the appendix.
>
> 3) According to your suggestions, we additionally conduct out-of-domain experiments according to your request. Since MFIDDR and DRTiD are the only two multi-view fundus datasets, we use the weights trained on MFIDDR and test comparative models on a four-view variant of DRTiD.
>
> **Table A7: Out-of-Domain DR Grading Performance Comparison on Four-View DRTiD (Trained on MFIDDR).**
>
> | Method   | Acc | F1 |
> |----------|-------------|------------|
> | MVCINN | 28.67       | 85.26      |
> |SMVDR| 78.55| 78.11|
> |WMIMVDR  | 78.04| 77.74 |
> |MVCBM   | 76.46 | 76.89  |
> |CEM (MV) | 78.62 | 77.32 |
> | Ours     | **79.31**   | **79.02**  |
>
> 4) As for the source and order of the clinical text and concept labels, we obtain the concept descriptions of DR lesions through a six-round dialogue with GPT-4. In each round, the model extracts specific feature information about a lesion, ultimately generating an accurate medical text description. The generated text includes the lesion's typical appearance, key features, and its manifestations at different stages of DR, as shown in **Fig. 10 of the revised manuscript**. The text is fed into a frozen text encoder text-embedding-3-large (TE3) to generate the textual concept embedding $\( \mathbf{t}_{j}\in \mathbb{R}^{n_t}\)$, where $j$ denotes the \( j \)-th concept. The concept prediction labels are listed: hard exudates (EX), hemorrhages (HE), microaneurysms (MA), soft exudates (SE), vitreous hemorrhage (VH), and vitreous opacity (VO).

---

> ### Author Response · Authors · 2025-11-28
> **Follow-up on our rebuttal**
>
> Dear Reviewer TMiz,
>
> We have provided our response to your initial review and revised the manuscript to address each of your concerns. With the discussion period ongoing, we would be grateful if you could spare a moment to review our rebuttal. We stand ready to answer any further questions or offer additional clarifications as required.
>
> Your insights will be invaluable in refining the final version of our work. Thank you once again for your careful time and valuable efforts!
>
> Thanks,
>
> Authors of submission 13076

---

### Author Response · Authors · 2025-11-27
**Thank you to all reviewers and AC**

We sincerely appreciate the time and effort invested by the reviewers in evaluating our manuscript, as well as their constructive and encouraging feedback. We are grateful for their recognition of our work’s contributions and the thoughtful comments that have helped us improve the paper.

Overall, according to the reviews, our summarization is as follows:

**Strengths:**

The framework design aligns with clinical logic, featuring interpretability and provenance. It adopts a workflow of "lesion concept detection → guideline-style reasoning → multi-view fusion", supports clinical intervention and result updates during testing, and is convenient for auditing and practical medical applications (Reviewer TMiz, Reviewer pmak, Reviewer eP2g).

The multi-view fusion strategy has theoretical support. The rationality of dynamic weight allocation is proven through generalization bounds, and the core logic is easy to understand (Reviewer TMiz, Reviewer pmak).

This model architecture design is innovative. It includes a lightweight Hilbert-RWKV backbone network that preserves locality and a vision-text alignment module (VT-RWKV), balancing efficiency and performance (Reviewer pmak, Reviewer Dsqb).

The performance of grading and concept prediction is better than that of baseline models. Ablation experiments verify the contribution of each module, and the inference efficiency is competitive (Reviewer TMiz, Reviewer pmak, Reviewer Dsqb, Reviewer eP2g).

**Weaknesses:**

**Experiments**: Key flaws lie in experiment design, verification, data, and practicality. Such as, uncertainty verification is insufficient, supervision leakage risk, no out-of-domain testing or robustness checks for low-quality views, and a lack of comparisons with hierarchical window Transformers like Swin.

**Presentation**: The main focus lies in the elaboration of the mechanism, the disclosure of details, and the aspect of explainability. A detailed explanation of the advantages of the Hilbert curve and the selection of the TE3 encoder is necessary. The key experimental details, including the acquisition of text prompts and concept labels, need to be disclosed. It is recommended to showcase the interpretability process more.

Based on the reviewers’ valuable suggestions and comments, we have conducted additional experiments and enhanced the clarity of the manuscript as requested. We believe these improvements have significantly strengthened the quality of our work. **The revised manuscript has been uploaded. The modified parts are marked in blue.**

Thank you again for your professional and careful review comments. Your encouragement and suggestions motivate us to continue contributing to the community.

If you have any further questions or concerns, we would be happy to address them. Your feedback will be highly appreciated.

---

### Note · Authors · 2026-01-26

I have read and agree with the venue's withdrawal policy on behalf of myself and my co-authors.

---

### Meta-Review · Area_Chair_HpYg · 2025-12-31

**Summary:**

The paper proposes a multi-view diabetic retinopathy (DR) diagnosis framework (ProConMV) utilizing a "Hilbert-RWKV" backbone and a dual uncertainty-aware fusion mechanism. The authors claim to enhance interpretability and accuracy by integrating lesion masks and clinical text prompts into the reasoning process. While the reviewers acknowledged the novelty of the backbone and the motivation for interpretability, the consensus for rejection stems from fundamental concerns regarding the validity of the "multimodal" setup and the theoretical grounding of the uncertainty mechanism.

A primary concern driving this decision is the artificial construction of the multimodal inputs. As pointed out by Reviewer Dsqb, the "provenance" and "multimodality" rely heavily on synthetic data—lesion masks generated by a pre-trained network and fixed, guideline-style text descriptions from GPT-4—rather than patient-specific clinical data. This raises significant doubts about whether the system is truly "reasoning" or merely performing feature engineering with frozen priors. Furthermore, while the authors provide empirical scatter plots to justify their uncertainty-based fusion, Reviewer TMiz correctly noted that rigorous calibration analysis (e.g., ECE/ACE) was missing, leaving the theoretical claims about generalization bounds insufficiently supported by the empirical evidence. The complexity of the proposed pipeline appears disproportionate to the marginal performance gains observed, particularly when the "multi-view" assumption in fundus photography is often inconsistent with standard clinical screening workflows.

**Reviewer Concerns:**

### **Addressed Concerns**

Backbone Comparison: The authors provided the requested comparisons with Swin Transformers and other locality-aware architectures, satisfying Reviewer pmak and Reviewer eP2g regarding the efficacy of the Hilbert-RWKV backbone.

Test-time Inputs: The authors clarified that only fundus images are required at inference time, with masks and text embeddings being derived or fixed.

Clarification of Mechanics: Questions regarding the specific generation of text prompts and the definition of the Hilbert curve mechanism were largely answered to the satisfaction of Reviewer eP2g.

### **Remaining Concerns**

Synthetic Multimodality & Provenance (Critical): Reviewer Dsqb’s major concern remains outstanding: the "multimodal" data is synthetically derived (fixed GPT text and pre-trained masks). The authors' rebuttal argues this is necessary due to data scarcity, but it fundamentally undermines the claim of "provenance-enabled" diagnosis if the provenance traces back to fixed templates rather than patient reality. This suggests the model is not truly multimodal in the clinical sense.

Uncertainty Calibration & Theory-Practice Gap: Reviewer TMiz requested rigorous calibration metrics (ECE, ACE) to validate the uncertainty mechanism. The authors provided scatter plots showing correlations between confidence and loss, but this is an empirical observation, not a rigorous proof of calibration. The theoretical claim that weighting by uncertainty reduces generalization error remains heuristically supported rather than rigorously proven in this specific implementation.

Clinical Validity of Interpretability: The "interpretability" relies on mapping inputs to fixed text descriptions. As noted by Reviewer Dsqb, without user studies or validation that these fixed descriptions actually aid clinical decision-making (vs. potentially biasing it), the claim of "human-machine collaboration" is weak.

**Reviewer Scores:**

The reviews reflect a significant divergence in evaluation, ranging from strong support to strong rejection. Reviewer pmak assigned an 8 (Accept), praising the principled fusion mechanism and the efficient Hilbert-RWKV backbone. Reviewer TMiz gave a 6 (Marginally Above Threshold), appreciating the provenance-aware pipeline but requesting more rigorous validation of the uncertainty calibration. Reviewer eP2g initially rated the paper a 4 (Marginally Below Threshold) due to concerns about clarity and backbone justification, but explicitly stated a recommendation to "accept" after the authors addressed their concerns during the rebuttal. In sharp contrast, Reviewer Dsqb maintained a 2 (Reject), arguing that the "multimodal" inputs were synthetically constructed (using frozen masks and GPT text) rather than clinically authentic, which they felt undermined the paper's core claims of validity and reproducibility.

---

### Decision · Program_Chairs · 2026-01-26

Reject